# Investigating the dynamics of microbial consortia in spatially structured environments

Sonali Gupta [1], Tyler D. Ross[1], Marcella M. Gomez[2], Job L. Grant[1], Philip A. Romero[1,3] & Ophelia S. Venturelli [1,3,4 ✉]

The spatial organization of microbial communities arises from a complex interplay of biotic and abiotic interactions, and is a major determinant of ecosystem functions. Here we design a microfluidic platform to investigate how the spatial arrangement of microbes impacts gene expression and growth. We elucidate key biochemical parameters that dictate the mapping between spatial positioning and gene expression patterns. We show that distance can establish a low-pass filter to periodic inputs and can enhance the fidelity of information processing. Positive and negative feedback can play disparate roles in the synchronization and robustness of a genetic oscillator distributed between two strains to spatial separation. Quantification of growth and metabolite release in an amino-acid auxotroph community demonstrates that the interaction network and stability of the community are highly sensitive to temporal perturbations and spatial arrangements. In sum, our microfluidic platform can quantify spatiotemporal parameters influencing diffusion-mediated interactions in microbial consortia.

[1] Department of Biochemistry, University of Wisconsin-Madison, Madison, WI 53706, USA. [2] Applied Mathematics, University of California Santa Cruz, Santa Cruz, CA 95064, USA. [3] Department of Chemical and Biological Engineering, University of Wisconsin-Madison, Madison, WI 53706, USA. [4] Department of Bacteriology, University of Wisconsin-Madison, Madison, WI 53706, USA. ✉email: venturelli@wisc.edu

Microbiomes ranging from soil[1] to the human gastro-intestinal tract[2] exhibit spatial organization spanning multiple scales: variation in abiotic parameters dictate behaviors over centimeters to meter, whereas inter-microbial interactions impact community behaviors over micrometers to centimeters[3,4]. The spatial structure of microbial communities has been shown to influence ecological stability, functional activities, and responses to environmental perturbations[5–9]. However, we do not fully understand the effects of microbial spatial distributions on community functions and stability, or how to manipulate spatial and temporal dynamics to program community properties.

The majority of microbial interactions are mediated by diffusible compounds[9], which can enhance or inhibit community member's growth rates, as well as modify the activities of intracellular networks. The spatial proximity of community members is a major determinant of the costs and benefits of microbial interactions, and shapes the evolution of ecological networks[10,11]. Spatial structure has been shown to provide ecological benefits, such as promoting population survival through local public good production[12] and enhancing biofilm resilience to environmental perturbations[7,13]. Spatial heterogeneity can also enable coexistence among members of a community by modulating the distribution of positive and negative interactions[5,14]. Finally, spatial structure can dictate the outcome of invasion of non-resident strains into a community[15].

There are key challenges to studying and controlling the spatial arrangement of microbes on the micrometer scale[3,4]. Bacterial populations have been physically separated using patterned agarose[16], hydrogels[17], partitioned microfluidics[18], nanoporous membranes[14], cellulose nanofibrils[19], nanochannels[20], and bio-printing[21] techniques to study interactions and chemical signal communication. We develop a microfluidic platform, MISTiC (Mapping Interactions across Space and Time in Communities), to deepen our understanding of the effects of defined spatial structure and temporally changing environmental signals on microbial community properties. MISTiC enables temporal control of environmental inputs, spatial control of bacterial populations on the micrometer-scale, and time-lapse imaging of single-cell and population-level growth and gene expression in a continuous culture environment.

We use MISTiC to quantify the role of defined spatial structure and fluctuating environmental signals on information transmission and the temporal robustness of a distributed gene circuit oscillator in synthetic microbial consortia. We demonstrate that spatial separation can enhance the fidelity of information transmission and biomolecular feedback loops can critically shape the stability of the oscillator to variations in spatial positioning. In addition, we investigate how spatial arrangements influence metabolite cross-feeding and community stability using an amino-acid auxotroph consortium. The inferred interaction networks in different environmental contexts within MISTiC coupled to measurements of amino acid release highlight key parameters that determine the stability of the consortium. Together, these data show that MISTiC can be used to precisely quantify the role of micrometer-scale spatial separation and temporal perturbations on microbial interaction networks, community functions and stability.

## Results
### A microfluidic platform to interrogate microbial interactions.
We designed MISTiC to study diffusion-mediated microbial interactions between pairs of strains, which are major drivers of multi-member community behaviors[22–24] (Fig. 1a). Our microfluidic design balances the pressures between growth chambers and prevents convective flow through 25, 50, 100, and 250 μm interaction channels, which spatially separate the strains[25]. Each $10 \times 50 \times 1$ μm growth chamber contains ~150 cells that are restricted to a monolayer for real-time quantification of gene expression and growth. The interaction channels are <0.5 μm tall and structurally supported by 0.5 μm pillars, which serve as a physical barrier for the cells, while permitting diffusion of biomolecules between growth chambers. As the population grows and divides, excess cells are washed away by continuous media flow, enabling long-term imaging.

The environmental conditions can be dynamically controlled for each strain using separate inlets. To characterize the molecular gradients established across interaction channels, we loaded a fluorescent dye into the source chambers and imaged its diffusion across the interaction channel into the sink chambers (Supplementary Fig. 1). The concentration of fluorescein within the interaction channels decreased as a function of the distance from the source chamber. The average fluorescein concentrations within sink chambers decreased with increasing interaction channel length. We used this data to develop a computational model that represents diffusion as a one-dimensional process by discretizing the interaction channel into 1 μm regions (Methods and Supplementary Methods). A linear degradation rate of fluorescein was required to recapitulate the steady-state experimental data (Supplementary Fig. 1).

### Investigating unidirectional bacterial signaling.
Microbes communicate via chemical signals to monitor their population size, coordinate gene expression, and efficiently allocate intracellular resources[26–29]. We investigated the impact of spatial separation on the dynamics of signal communication mediated by quorum-sensing between engineered *Escherichia coli* populations. This community consisted of a sender strain that produced a quorum-sensing signal (3-oxo-C6-HSL or AHL) and a receiver strain that sensed the signal and activated a red fluorescent reporter (RFP) (Fig. 1b). The sender strain harbored an arabinose-inducible AHL synthetase (LuxI) transcriptionally fused to a green fluorescent protein (GFP) and the receiver strain contained an aTc-inducible AHL receptor (LuxR) and RFP regulated by a LuxR promoter (Supplementary Fig. 2).

We seeded MISTiC growth chambers with the sender and receiver strains and monitored their gene expression using time-lapse fluorescence microscopy (Methods) (Experiment 1, Table 1). After an initial growth phase, arabinose was administered to induce inter-strain communication. The steady-state of GFP was constant across distance, demonstrating the uniformity of arabinose concentration across the device (Fig. 1c). By contrast, the receiver's steady-state RFP expression decreased as a function of distance from the sender strain (Fig. 1d). The RFP response time did not vary with the length of the interaction channel (Supplementary Fig. 3a). The orientation of the sender and receiver strains within the growth chambers did not impact the gene expression patterns (Experiment 2, Table 1, and Supplementary Fig. 4a, b), indicating the absence of pressure imbalances and convective flow within MISTiC. In sum, these results indicate that MISTiC can resolve quantitative changes in the spatiotemporal patterns of gene expression in microbial consortia.

### A dynamic model for inter-strain communication.
To understand the roles of molecular factors in diffusion-mediated inter-strain communication within MISTiC, we constructed a dynamic computational gene expression model building on our chemical diffusion model (Fig. 2a and Supplementary Methods). In the sender strain, the model captures the concentrations of arabinose (*ara*), GFP mRNA ($GFP_m$), GFP protein ($GFP_p$), LuxI mRNA

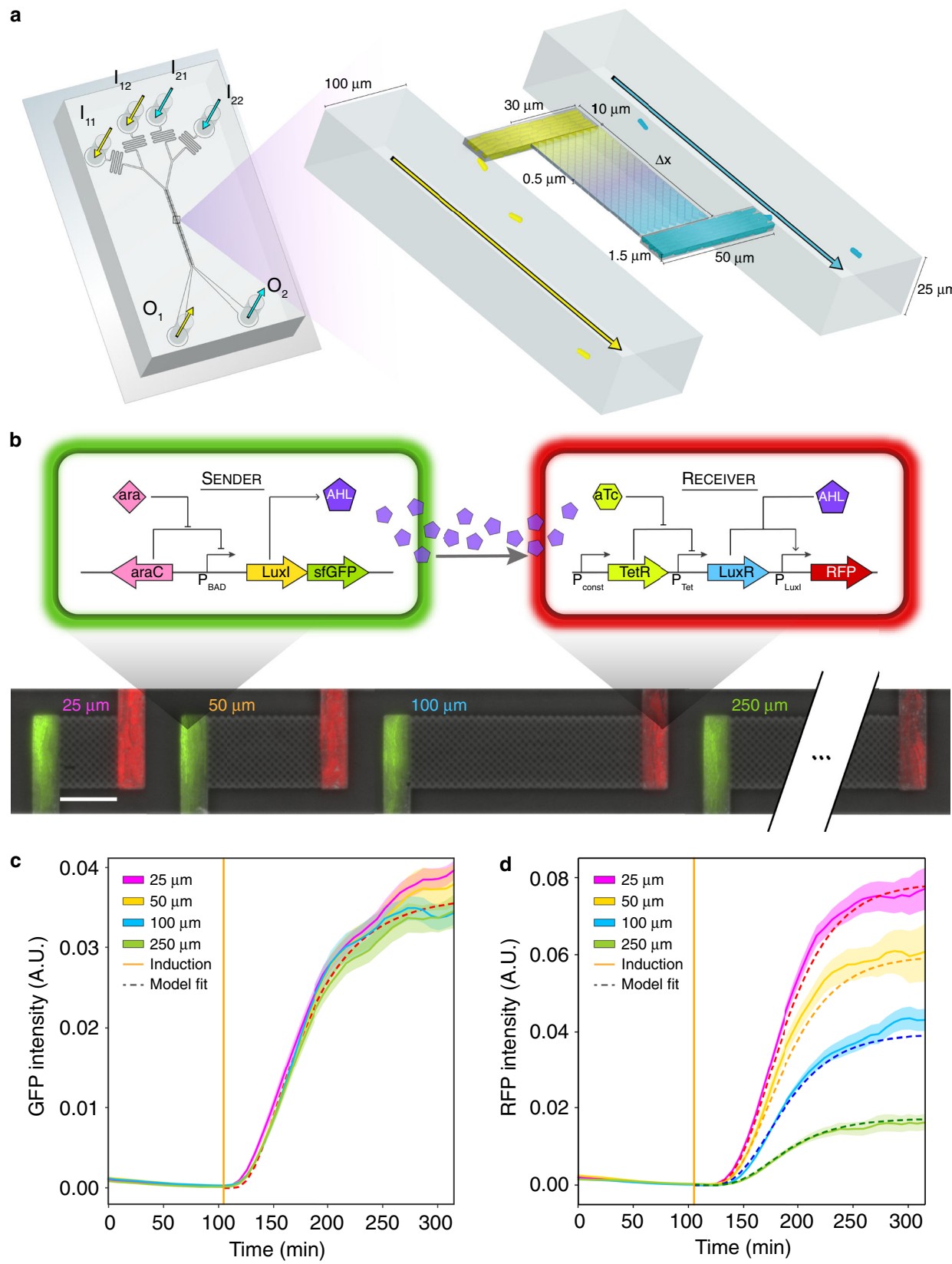

(*LuxI*$_\text{m}$), LuxI protein (*LuxI*$_\text{p}$), and AHL (*AHL*). The species in the receiver strain include RFP mRNA (*RFP*$_\text{m}$), RFP protein (*RFP*$_\text{p}$), LuxR protein (*LuxR*$_\text{tot}$), and the activated receptor (*LuxRAHL*) consisting of a complex of AHL bound to LuxR. The model includes time delays for arabinose transport and the sequential assembly reactions of *GFP*$_\text{p}$, *LuxI*$_\text{p}$, and *RFP*$_\text{p}$[30]. We used a genetic algorithm to estimate the parameters based on time-series fluorescent reporter measurements (Methods).

The model accurately recapitulates the temporal changes in GFP and RFP at different distances (Fig. 1c, d). The model

**Fig. 1 Design of a microfluidic platform to investigate the role of spatiotemporal parameters in microbial consortia. a** Schematic of the microfluidic device. The inlets $I_{11}$, $I_{12}$ or $I_{21}$, $I_{22}$ connect to the outlets $O_1$ or $O_2$, respectively, and allow temporal control of the environmental conditions. Cells are initially seeded into growth chambers and the continuous flow of media through the main channels removes excess cells. Pairs of growth chambers are separated by a lattice of pillars, defined as the interaction channel, which allows diffusion of biomolecules and prevents cells from entering the interaction channel. The device has ten pairs of growth chambers for each separation distance. **b** Top: schematic of the genetic circuit in the *E. coli* sender and receiver strains. In the sender strain, the operon containing the synthetase LuxI and GFP is induced in response to arabinose. The synthetase LuxI produces the acyl-homoserine lactone (3-oxo-C6-HSL or AHL), which diffuses through the interaction channel into the receiver strain growth chamber. In the receiver strain, AHL binds to LuxR to form an activated LuxR–AHL complex, which in turn activates expression of RFP driven by a LuxR-regulated promoter. Bottom: overlaid representative fluorescence and phase-contrast microscope images of the sender and receiver strains in the device for each interaction channel length. The scale bar represents 25 μm. **c** GFP fluorescence in sender growth chambers as a function of time. The vertical line indicates the time at which arabinose was introduced. Shaded regions represent one standard deviation from the mean. The dashed line denotes the model fit. **d** RFP fluorescence over time in the receiver growth chambers. The vertical line indicates the time at which arabinose was introduced. Shaded regions represent one standard deviation from the mean. The dashed line denotes the model fit. Source data are provided as a Source Data file.

predicts that the receiver steady-states are highly sensitive to variation in spatial separation less than 100 μm and forecasts that ~150 sender cells can transmit information across 1000 μm (defined as 1% of the $RFP_\mathrm{p}$ steady-state for 2 μm separation) (Fig. 2b). In the model, increasing the separation distance from 25 to 250 μm resulted in a 2.3 min RFP response time delay (Supplementary Fig. 3b), consistent with an unresolved time delay in our experimental data (Supplementary Fig. 3a).

The diffusion rate of AHL into the main channel ($D_2$) and the degradation rate of AHL ($\gamma_\mathrm{AHL}$) influence the AHL concentration gradient established in the interaction channel. We sought to investigate the effects of these parameters on the distance-dependent gene expression pattern. Setting $\gamma_\mathrm{AHL}$ and/or $D_2$ to zero significantly altered absolute $RFP_\mathrm{p}$ steady-state concentrations and their relative differences across distance, indicating that the stability of the chemical signal and the physical properties of the environment can dictate the response of a microbial community to spatial separation (Supplementary Fig. 5).

In response to temporally changing environmental stimuli, the allocation of intracellular resources can be optimized by modulating the response times of intracellular networks[31,32]. Therefore, we explored how the $RFP_\mathrm{p}$ response time depends on two key parameters: the diffusion constant through the interaction channel, $D_1$, and the binding affinity of $LuxRAHL$ to the RFP promoter, $K_\mathrm{RFP}$. The delay in $RFP_\mathrm{p}$ increases with decreasing $D_1$ and remains relatively constant as a function of $K_\mathrm{RFP}$. At intermediate values of $D_1$, the delay is inversely related to $K_\mathrm{RFP}$ (Fig. 2c). The estimated parameters for the sender–receiver consortium map to a regime that display small time delays, indicating that a measurable time delay between the 25 and 250 μm conditions would require a large change in $D_1$.

We analyzed the effects of biochemical parameters on the relative changes in steady-state $RFP_\mathrm{p}$ as a function of distance, defined as distance sensitivity. Changing the binding affinity of LuxR to AHL ($K_\mathrm{LuxR}$) and/or $K_\mathrm{RFP}$ (Fig. 2d), or the Hill coefficient of $RFP_\mathrm{p}$ production ($n_\mathrm{RFP}$) (Fig. 2e) relative to their estimated values shifts the system between the linear and saturated regimes of the $AHL$ dose response. In the linear regime, steady-state $RFP_\mathrm{p}$ exhibits larger relative differences as a function of distance (greater distance sensitivity) compared with the saturated regimes (lower distance sensitivity). These results suggest that circuits could be programmed to realize different spatial patterns by modifying distance sensitivity via ultrasensitivity[33,34], the affinity of transcription factors to promoters or a chemical inducer[35], or the concentration of molecular factors in the circuit (Fig. 2f).

**Parameters impacting fidelity of information transmission.** The spatial organization of a community can influence its response to temporal variations in environmental stimuli[36]. We applied a periodic input to the sender–receiver consortium to

characterize how distance impacts information transmission in fluctuating environments. To predict system behavior, we simulated square wave oscillations in arabinose (*ara*) with a period of 2 h (Fig. 3a). The steady-state amplitude and mean $RFP_\mathrm{p}$ decreases with increasing spatial separation. To test the model predictions, the sender–receiver consortium was exposed to alternating arabinose between 0% and 0.1% with a period of 2 h (Supplementary Movie 1) (Experiment 3, Table 1). The periods of GFP and RFP were synchronized with the arabinose input and therefore did not vary with distance (Supplementary Fig. 6a). In response to the oscillatory signal, both GFP and RFP mean intensities increased over time and reached a steady-state oscillatory phase (Fig. 3b, c). The GFP mean and amplitude did not vary across distance (Fig. 3b and Supplementary Fig. 6b). Mirroring the model prediction, the RFP mean and amplitude decreased as a function of distance (Fig. 3b, c and Supplementary Fig. 6b). At steady-state, the mean activation and decay response times of GFP and RFP were ~30–40 min across all distances (Supplementary Fig. 6c, d). These response times are similar to the exponential phase doubling time of *E. coli* in Luria Broth (LB) media, suggesting that cell growth/division dictated the oscillatory timescale.

We next explored whether information transfer across distance is corrupted by stochastic processes within the cell[37] near a critical input frequency. We measured the response of the sender–receiver consortium to a periodic arabinose input with a period of 1 h (Experiment 4, Table 1). Simulations of arabinose oscillations with a 1 h period yielded a distance-dependent change in steady-state $RFP_p$ with lower means and amplitudes than the response to an input period of 2 h (Fig. 3a, d). In response to alternating arabinose concentrations with a 1 h period, GFP exhibited steady-state oscillations and the GFP mean did not vary across distance (Fig. 3e). RFP displayed irregular variability around the mean at steady-state and a distance-dependent change in the steady-state fluorescence intensity (Fig. 3f).

Distance or higher input frequencies reduced the RFP amplitude in model simulations and in our experimental data (Fig. 3g). These results demonstrate that distance establishes a low-pass filter, wherein the RFP amplitude above a threshold ($A_c$) will vanish at high frequencies and larger distances. Increasing $n_\mathrm{RFP}$ augments the relative differences in RFP amplitudes across distance as a function of the input period (Supplementary Fig. 7a), indicating that ultrasensitivity can establish a switch-like low-pass filter with distance. In the 1 h forced oscillator experiment, the RFP amplitudes did not vary across distance (Fig. 3f, g), highlighting diminished information transmission in diffusion-mediated communication[38].

Communication between physically separated populations is impacted by extracellular noise due to diffusion[39] and noise from intracellular processes such as transcriptional[40] or translational

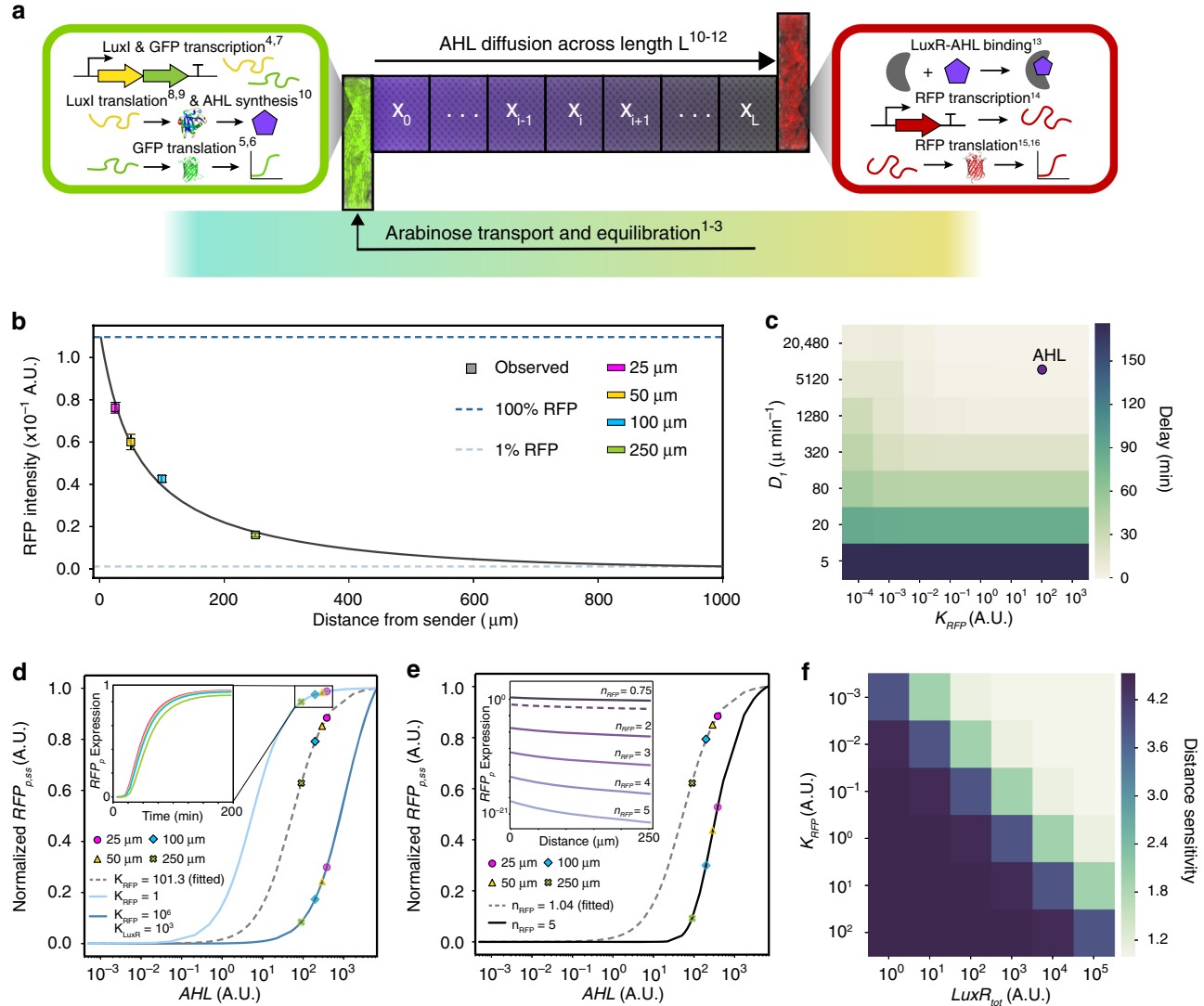

**Fig. 2 Computational model of inter-strain communication in defined spatial environments. a** Model schematic depicting the physical and biological processes represented by the model equations (Supplementary Methods). **b** The model $RFP_p$ steady-states decrease with the distance from the sender strain. The blue dashed line represents the maximum $RFP_p$ steady-state concentration for a 2 μm interaction channel. The gray dashed line denotes 1% of the maximum steady-state $RFP_p$ concentration. Data points (squares) represent experimental measurements and error bars denote 1 SD from the mean. **c** Heat map of the simulated $RFP_p$ time delays for diffusible signal molecules spanning a broad range of diffusion rates ($D_1$) and binding affinities of $LuxRAHL$ to the $RFP_p$ promoter ($K_{RFP}$). The circle represents the estimated parameters based on experimental data. **d** $RFP_p$ steady-state dose response as a function of AHL for different $K_{RFP}$ and $K_{LuxR}$ values. The dashed curve represents the dose response of the parameterized model based on the experimental data. Inset denotes representative simulations of $RFP_p$ for different interaction channel lengths. The $RFP_p$ steady states for each interaction channel length (colored marker styles) were computed for $ara = 10$. **e** Steady-state $RFP_p$ dose response as a function of $AHL$ for two different $n_{RFP}$ values in the model. The dashed curve represents the dose response for the parameterized model based on the experimental data. The $RFP_p$ steady-states for each interaction channel length (colored marker styles) were computed for $ara = 10$. Inset: relationship between distance and steady-state $RFP_p$ concentration for a range of $n_{RFP}$ values. **f** Heatmap of distance sensitivity across a broad range of total LuxR concentrations ($LuxR_{tot}$) and binding affinities of the activated LuxR complex to the $RFP_p$ promoter ($K_{RFP}$). Distance sensitivity is defined as the ratio of steady-state $RFP_p$ concentration in a 25–250 μm condition. Source data are provided as a Source Data file.

bursting[41]. We investigated how information in the periodic input signal is encoded in the frequency domains of the gene expression responses. The power spectrum represents how the variance in gene expression is distributed across frequencies. The power spectrum for GFP and RFP displayed prominent peaks at the frequency of the input signal for both experiments (Fig. 3h and Supplementary Fig. 7b, c). The major peak in the RFP power spectrum at the input frequency decreased with distance in the 2 h forced oscillation experiment (Fig. 3h, top), reflecting the trend in amplitude across distance. In response to a 1 h period, RFP

power spectrum frequencies larger than the signal bandwidth decreased with distance (Fig. 3h, bottom).

To evaluate the fidelity of information transmission across distance, we defined the signal-to-noise ratio (SNR) as the total power of the input signal bandwidth divided by the total power across all frequencies greater than the signal bandwidth (Methods). The GFP SNR did not vary as a function of distance in either of the forced oscillator experiments (Fig. 3i and Supplementary Fig. 7d). In response to a 2 h period, the RFP SNR was inversely related to distance since it was dominated by

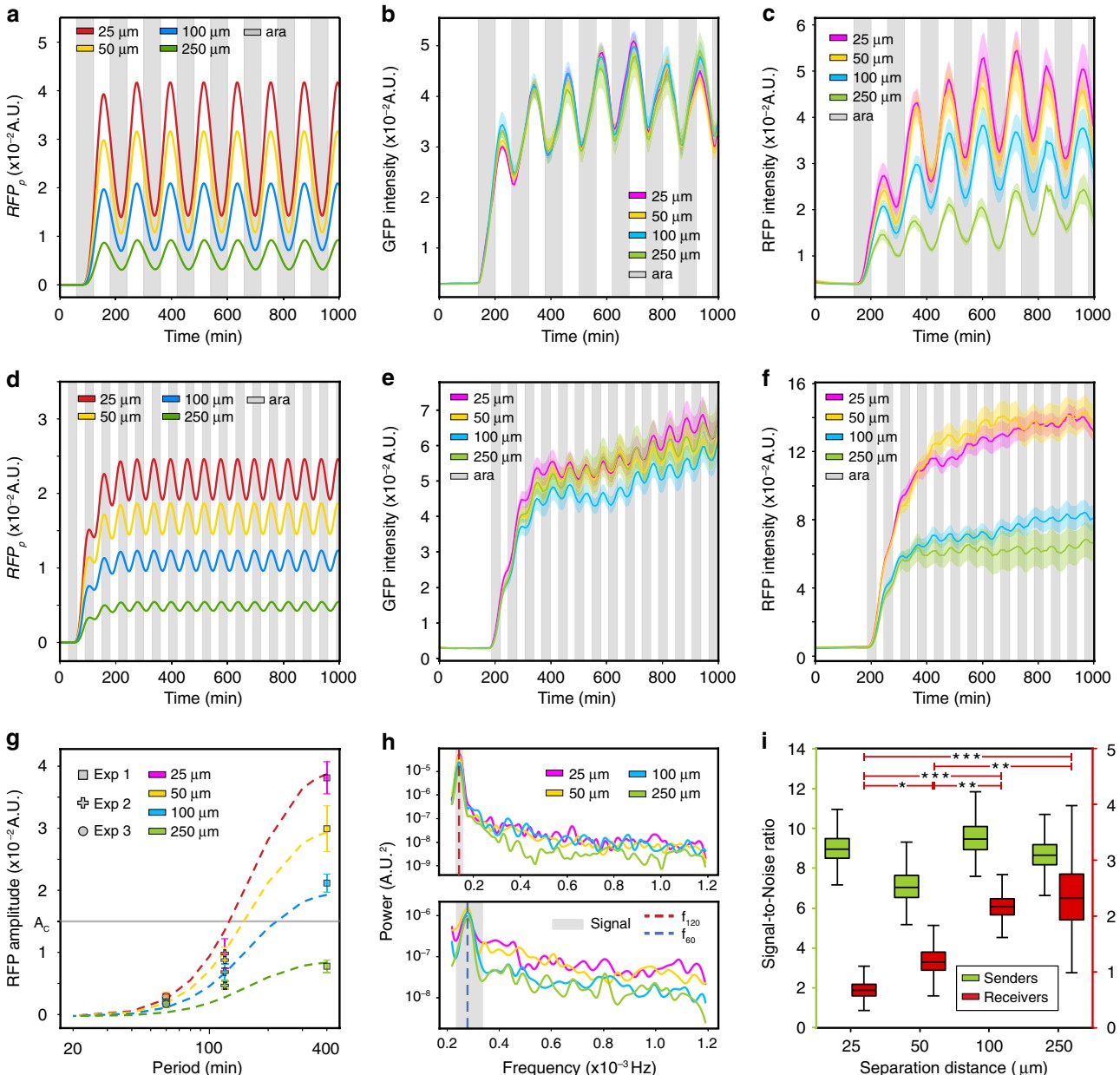

**Fig. 3 The frequency of a periodic input determines the effect of distance on information transmission. a** Model simulations of $RFP_p$ in response to a periodic arabinose input (2 h period). Gray shaded regions denote the presence of arabinose. **b** GFP as a function of time in response to an oscillatory arabinose input (2 h period). Gray shaded regions denote the presence of 0.1% arabinose. Colored shaded regions represent 1 SD from the mean. **c** RFP as a function of time in response to a periodic arabinose input (2 h period). **d** Model simulations of $RFP_p$ in response to a periodic arabinose input (1 h period). **e** GFP over time in response to an oscillatory arabinose input (1 h period). **f** RFP in response to an oscillating arabinose input (1 h period). **g** RFP amplitude as a function of the input period at different distances. Square, plus, and circular data points represent the amplitudes in the step response, 2 h and 1 h forced oscillator experiments, respectively. Error bars represent 1 SD from the mean amplitude. **h** Top: mean RFP power spectra in response to an input period of 2 h (red dashed line) for 395–1536 min. The gray shaded region denotes a bandwidth of ±10 min around the expected input frequency. Bottom: mean RFP power spectrum in response to an input period of 1 h (blue dashed line) for 500–1249 min. **i** Signal-to-noise (SNR) ratios computed using bootstrapped power spectra for GFP or RFP for each distance (Methods) in the 1 h forced oscillator experiment. The horizontal lines within boxes denote the median and upper and lower edges represent the upper and lower quartiles, respectively. Upper and lower whiskers represent the 95th and 5th confidence intervals, respectively. Horizontal lines with stars denote statistically significant differences ($P < 0.05$) based on a one-tailed bootstrapped hypothesis test (Methods). $*P < 0.05$, $**P < 0.01$, and $***P < 0.001$ (Supplementary Fig. 16). Source data are provided as a Source Data file.

the power of the input signal bandwidth (Supplementary Fig. 7d). Notably, the RFP SNR increased with distance in the 1 h forced oscillation experiment (Fig. 3i).

Therefore, in the regime of a critical input frequency[38], the fidelity of inter-strain communication is diminished at short distances due to elevated noise but diffusible signals have a

limited spatial range over long distances[42]. Our results suggest that the reliability of information transmission in bacterial signal communication can vary non-monotonically with distance in response to inputs near a critical frequency. Together, these data show that distance can function as a low-pass filter to allow cells to selectively respond to prolonged environmental fluctuations.

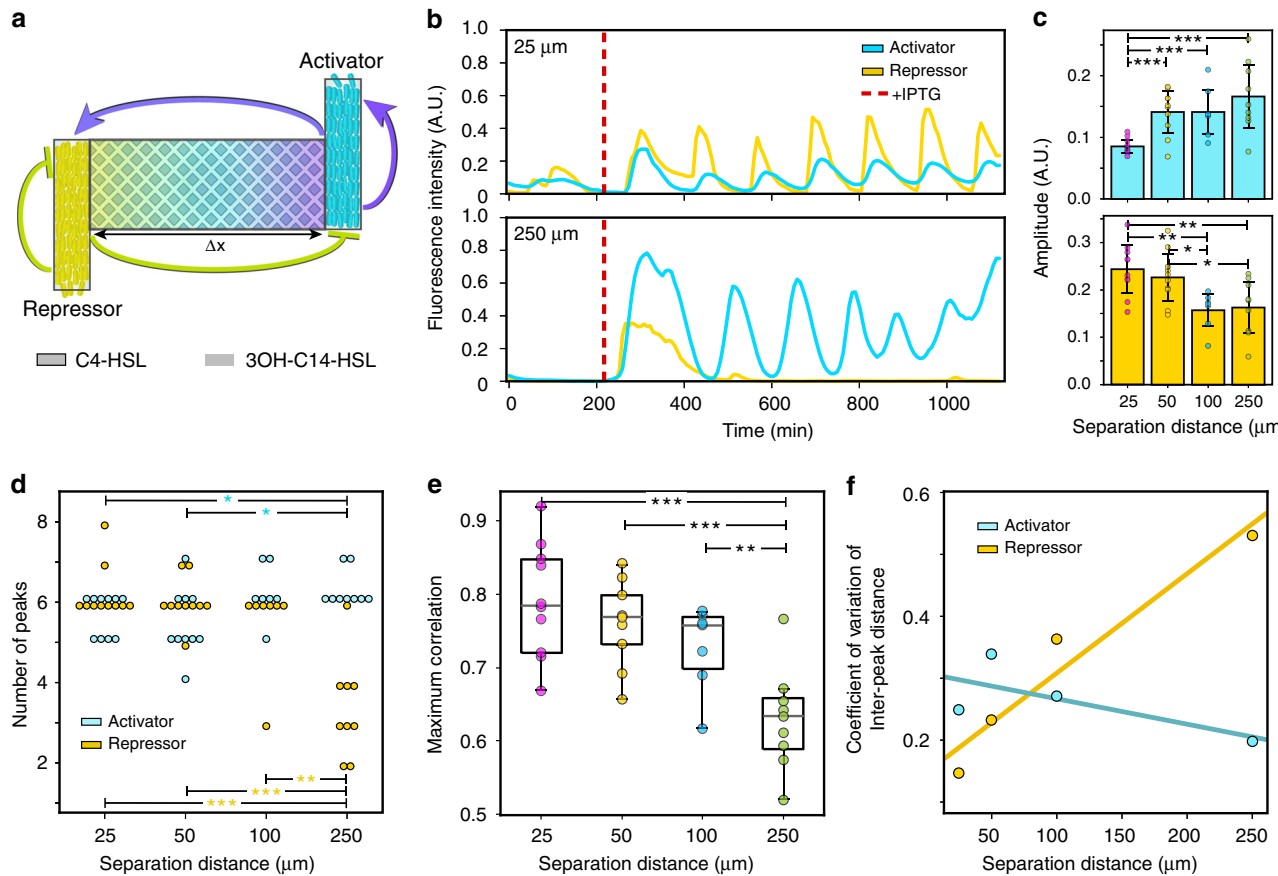

**Fig. 4 Feedback loops dictate the role of distance on a distributed gene circuit oscillator. a** Network schematic of activator and repressor strains in the MISTiC device. The activator and repressor exhibit bidirectional communication and dual-feedback loops mediated by the signaling molecules C4-HSL and 3OH-C14-HSL. The activator and repressor exhibit positive and negative feedback loops, respectively. **b** Representative normalized CFP and YFP fluorescence intensities as a function of time in the 25 μm (top) and 250 μm (bottom) conditions. The red line indicates the time of induction with 1 mM IPTG. **c** Amplitude of peaks in the mean-subtracted CFP (top) and YFP (bottom) fluorescence intensities across distances. Error bars represent 1 SD from the mean. Horizontal lines with stars denote a statistically significant difference ($P < 0.05$) based on a two-sided $t$-test. *$P < 0.05$, **$P < 0.01$, ***$P < 0.001$ (Supplementary Fig. 16). **d** The number of peaks as a function of distance for activator and repressor strains. Horizontal lines denote a statistically significant difference ($P < 0.05$) based on a two-sided $t$-test. *$P < 0.05$, **$P < 0.01$, and ***$P < 0.001$ (Supplementary Fig. 16). **e** Maximum cross-correlation between paired CFP and YFP time-series measurements for each distance. The horizontal lines within boxes denote the median and upper and lower edges represent the upper and lower quartiles, respectively. Upper and lower whiskers represent the 95th and 5th confidence intervals, respectively. Horizontal lines with stars denote a statistically significant difference ($P < 0.05$) based on a two-sided $t$-test. *$P < 0.05$, **$P < 0.01$, and ***$P < 0.001$ (Supplementary Fig. 16). **f** The coefficient of variation of the inter-peak distances (phase drift) as a function of separation distance for activator and repressor strains. Lines represent linear regression fits to the data. Source data are provided as a Source Data file.

Above a critical input frequency where cellular noise dominates, spatial separation can modify a trade-off between the reliability of information transmission and the magnitude of the output response.

**Feedback loops impact oscillatory dynamics across distance.** The intracellular networks mediating microbial interactions can comprise interlinked feedback loops and bidirectional communication. We sought to understand the effects of spatial separation on the dynamics of a distributed gene circuit oscillator consisting of an *E. coli* activator strain that produces C4-homoserine lactone (C4-HSL), which induces the enzymatic synthesis of 3-OHC14-HSL in an *E. coli* repressor strain (positive inter-strain interaction)[43]. The activator displays a positive feedback loop by autoregulating the circuit controlling C4-HSL production (Fig. 4a). The signal 3-OHC14-HSL produced by the repressor strain induces the expression of a quorum-quenching lactonase *aiiA* in the activator and repressor strains, which degrades both signals and thus inhibits circuit activity in the repressor (negative feedback loop) and activator (negative inter-

strain interaction). Identical promoters driving the synthetases in the activator and repressor strains also regulated the expression of cyan fluorescent protein (CFP) and yellow fluorescent protein (YFP), respectively, to monitor circuit activity in real time.

The reporters CFP and YFP displayed oscillations across the majority of conditions in the MISTiC device (Fig. 4b, Supplementary Fig. 8, and Supplementary Movie 2) (Experiment 5, Table 1). Paired growth chambers exhibited synchronized oscillations whereas unpaired growth chambers were not synchronized, indicating that signal diffusion through the interaction channels was critical to the temporal coordination of gene expression. The CFP amplitude increased with distance whereas the YFP amplitude displayed the reverse trend (Fig. 4c), signifying reduced synchronization of gene expression with distance. The number of activator peaks moderately increased and number of repressor peaks substantially decreased with distance (Fig. 4d), indicating that the oscillatory behavior of the repressor was highly sensitive to variations in the interaction range. Our data showed that the activator amplitudes varied non-monotonically with distance: the amplitudes were diminished by

stronger repression at short distances, increased at intermediate distances, and disappeared at long distances as the activator expression approached a constitutive ON state (Supplementary Fig. 8). Conversely, the repressor amplitudes decreased with distance and displayed an abrupt loss of oscillatory behavior at a spatial separation of 100–250 μm from the activator strain.

The maximum cross-correlation between CFP and YFP decreased (Fig. 4e), and the time lag increased (Supplementary Fig. 9a) with the length of the interaction channel, showing that distance diminished the coordination of gene expression dynamics. The distribution of inter-peak distances quantifies the variability in oscillatory behaviors and is an indicator of phase drift[44]. The coefficient of variation of the inter-peak distance distribution increased by more than threefold for the repressor strain but did not change significantly for the activator strain (Fig. 4f and Supplementary Fig. 9b, c). In sum, the activator's positive feedback loop enhanced oscillatory robustness whereas the repressor's negative feedback loop resulted in greater sensitivity to spatial separation.

**Spatial and temporal modes of metabolic interactions.** Microbial community functions are driven by metabolite-mediated interactions including competition over limiting resources, toxin release and cross-feeding[3,7,9,22,24,45]. To investigate how spatial separation influences metabolic interactions in microbial communities, we studied a synthetic *E. coli* consortium composed of a phenylalanine (Δ*pheA*) and methionine (Δ*metA*) auxotroph strain. Phenylalanine (F) and methionine (M) auxotrophies are predicted to be prevalent in microbial communities and are two of the most energetically costly amino acids to synthesize in *E. coli*[46].

We characterized the community dynamics in batch culture by inoculating the strains at three initial ratios (50% Δ*metA*, 50% Δ*pheA* or 90% Δ*metA*, 10% Δ*pheA*, or the reciprocal), in minimal media lacking M and F, and performed periodic transfers of the community to fresh media (Methods). Irrespective of the initial strain proportion, the co-culture converged to a Δ*pheA* dominated steady-state (Supplementary Fig. 10a). For a sustained 24 h transfer time, the community exhibited a decreasing trend in OD600 and eventually collapsed after the third passage (Supplementary Fig. 10b, top). However, community growth was maintained over three passages by switching the final passage to a 48 h incubation time, suggesting that a critical cell density was required to achieve ecological stability (Supplementary Fig. 10b, bottom). Therefore, in a batch culture environment that preserves a critical population size, the community exhibited growth and stable coexistence over multiple passages with Δ*pheA* dominating the community.

In MISTiC, the population-level growth rate can be inferred by the rate of dilution of an inducible and stable fluorescent reporter due to cell growth[37,47]. We used this method to determine each strain's maximum growth rate in a spatially structured and continuous culture environment by labeling Δ*metA* and Δ*pheA* with inducible GFP and RFP, respectively (Fig. 5a) (Methods). An interaction was quantified as the fold change of each strain's maximum growth rate in the 25 to 250 μm condition. Microbial interactions are frequently deciphered by evaluating the difference of a strain's growth parameters in monoculture and co-culture conditions[22,48]. In MISTiC, the sign and strength of the interaction could be deduced as a function of spatial separation. In the presence of all amino acids, Δ*metA* and Δ*pheA* exhibited similar average doubling times of 80 and 86 min (Experiment 6, Table 1), respectively, across all interaction channel lengths (Fig. 5b, c and Supplementary Fig. 11), indicating a neutral interaction network (Fig. 5h).

In the absence of M and F, Δ*metA* grew very slowly (1128 min doubling time) across all distances (Experiment 7, Table 1, Fig. 5b, and Supplementary Fig. 12a, c). By contrast, the doubling time for Δ*pheA* substantially increased with distance (Fig. 5c and Supplementary Fig. 12b, d). Therefore, the growth rate of Δ*pheA* was highly sensitive to distance from the Δ*metA* strain and not the reciprocal, highlighting a major difference in the strength and operating regime of the interactions (Fig. 5h, center). The maximum growth rate of Δ*pheA* was delayed by 43 min in the 25 to 250 μm condition (Fig. 5c, inset), demonstrating that the timing of the transition from lag phase to growth was also distance-dependent.

We next investigated the growth of Δ*pheA* and Δ*metA* in mixed conditions for comparison to the spatially separated context. Equal proportions of Δ*metA* and Δ*pheA* were introduced into the growth chambers and cultured in the absence of M and F (Experiment 8, Table 1). Single-cell segmentation and tracking were performed to distinguish the strains within mixed communities and quantify single-cell growth rates (Methods) (Supplementary Fig. 13a). The interaction channel length did not contribute to the variation in the growth rates of Δ*metA* and Δ*pheA* (Supplementary Fig. 13b, c). The average Δ*metA* and Δ*pheA* growth rates within a chamber decreased over time but remained non-zero for the majority of the experiment (Fig. 5d). In addition, the percentage of growing cells across all growth chambers in the mixed condition was larger than 40% for both strains for the majority of the experiment (Fig. 5d, inset). To evaluate the difference in each strain's growth rate in the co-culture compared to monoculture, similar MISTiC experiments were performed for Δ*metA* (Experiment 9, Table 1) and Δ*pheA* (Experiment 10, Table 1). The co-culture growth rates were significantly higher than their respective monoculture conditions, demonstrating a mutual growth benefit in mixed conditions within MISTiC.

The Spearman's correlation between the growth rate of each strain and the fraction of the growth chamber occupied by the partner strain was used to quantify the effect of the partner strain's abundance on growth rate (Fig. 5e, inset). Both strains exhibited statistically significant and non-zero Spearman correlations for the majority of the experiment, indicating a mutualism. The Spearman's correlation was consistently higher for Δ*pheA*, indicating a stronger dependence of the growth rate of Δ*pheA* on the abundance of Δ*metA* than the reciprocal, consistent with the physically separated experiment (Fig. 5b, c). In spite of the growth rate difference, the average ratio of the two strains approached a stable steady-state (Supplementary Fig. 13d). As the percentage of dividing cells (Supplementary Fig. 13e) was significantly lower than the percentage of growing cells (Fig. 5d, inset) for both strains, cell elongation was the dominant mode of growth in these conditions. In addition, growth rates did not vary as a function of the position of single cells within the growth chambers (Supplementary Fig. 13f). These findings illustrate the ability of MISTiC to resolve subpopulation growth heterogeneities based on single-cell data.

To determine how rescuing the growth of Δ*metA* impacted the interaction network, we examined the growth rates of the spatially separated strains in the presence of all amino acids except F (Experiment 11, Table 1). The doubling time of Δ*metA* was similar to its doubling time in the presence of M and F (Experiment 6, Table 1) across all distances (Fig. 5f and Supplementary Fig. 14a, d). The rate of change of RFP fluorescence was biphasic, indicating that Δ*pheA* had two growth modes in this condition (Methods and Supplementary Fig. 14b, c). The Δ*pheA* doubling times in the first growth phase did not change across distance, yielding a neutral interaction network (Fig. 5h and Supplementary Fig. 14e). The second Δ*pheA* growth

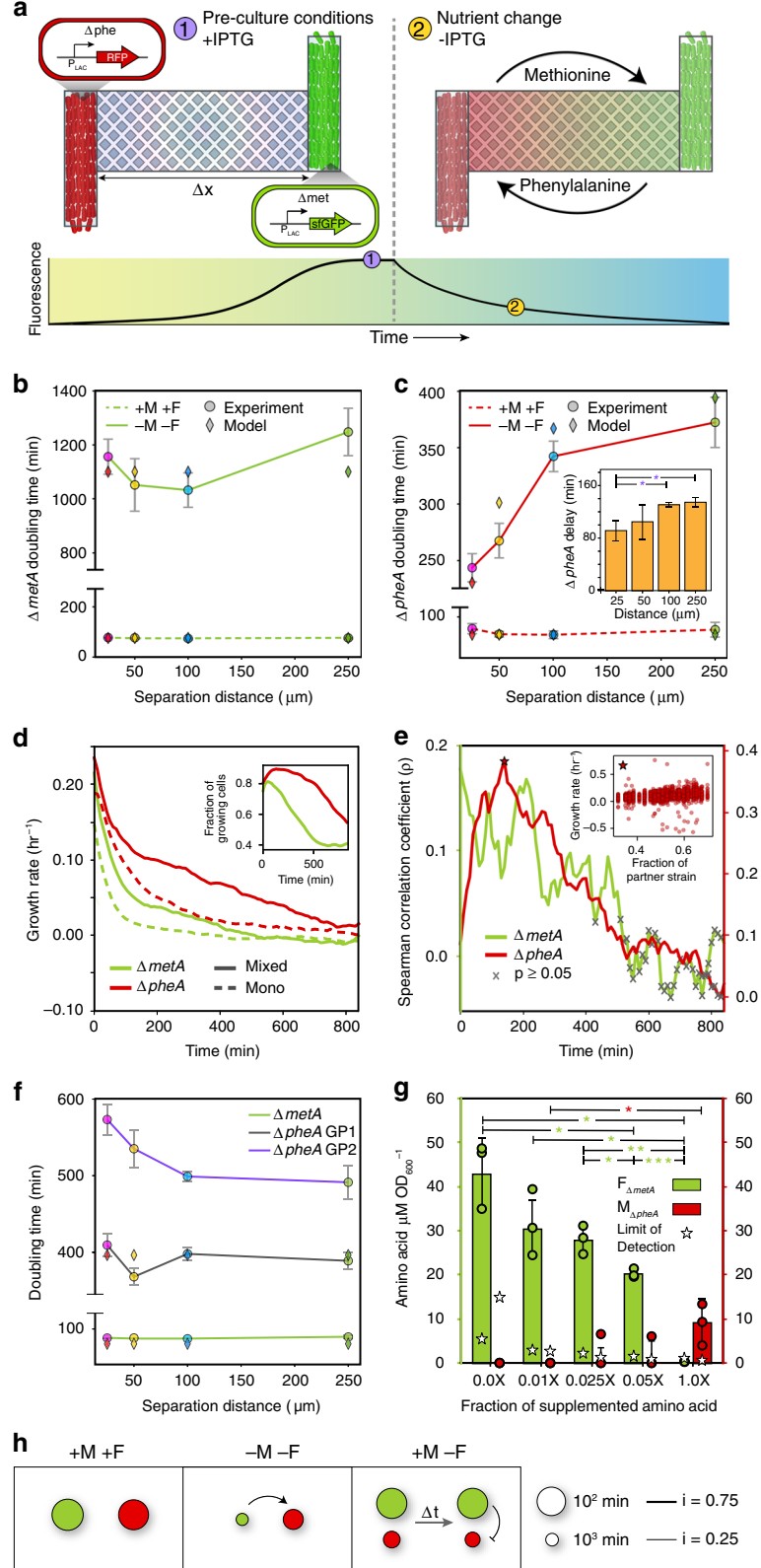

phase displayed a moderately competitive growth trend across a tenfold change in distance (Fig. 5f, h and Supplementary Fig. 14f). The positive interaction from *ΔmetA* to *ΔpheA* was abolished by rescuing *ΔmetA* growth, suggesting that rapid growth of *ΔmetA* resulted in a substantial decrease in the release rate of F.

To further study the inverse relationship between the growth rate of *ΔmetA* and the strength of its outgoing positive interaction

on *ΔpheA*, we measured M and F in each producer strain's supernatant across different concentrations of the rescuing amino acids (Methods). The F concentration per unit biomass in the *ΔmetA* supernatant was inversely proportional to the supplemented M concentration and thus its growth rate, consistent with the significant growth enhancement of *ΔpheA* when *ΔmetA* was slowly growing and metabolically active but not when it was

**Fig. 5 Spatial and temporal modes of amino acid cross-feeding in a synthetic *E. coli* consortium. a** Schematic of the experimental design. Δ*metA* and Δ*pheA* are labeled with IPTG-inducible fluorescent reporters. The strains are initially cultured in the presence IPTG. At the denoted time, IPTG is removed and the media condition is altered. The maximum growth rate is inferred based on the rate of decay of the fluorescence signal. Relationship between distance and population-level doubling times of (**b**) Δ*metA* or (**c**) Δ*pheA*. The dashed and solid lines represent presence and absence of methionine (M) and phenylalanine (F), respectively. Diamonds represent the model fits. Inset in **c**: the time of maximum Δ*pheA* growth rate in media lacking M and F. Horizontal lines denote a statistically significant difference ($P < 0.05$) based on one-tailed bootstrap hypothesis testing (Methods). *$P < 0.05$, **$P < 0.01$, and ***$P < 0.001$ (Supplementary Fig. 17). **d** Relationship between time and the average single-cell growth rates of Δ*metA* and Δ*pheA* in a mixed community (solid line) or monoculture (dashed line) in media lacking M and F. Inset: fraction of cells with non-zero growth rates as a function of time. **e** Spearman's correlation coefficient as a function of time between the fraction of the growth chamber occupied by the partner strain and the growth rate for individual cells (inset). The X symbol denotes correlations corresponding to $P > 0.05$. **f** Relationship between distance and the minimum doubling time in the presence of M and absence of F. Δ*pheA* exhibited two growth phases following the media switch (GP1 and GP2). Diamonds indicate model fits to the maximum growth rates in GP1. **g** Concentration of M or F divided by absorbance at 600 nm (OD600) in Δ*pheA* or Δ*metA* conditioned media. Stars indicate the limit of detection for each measurement. 1× Amino acid fraction refers to 0.2 mM M and 0.4 mM F, respectively. The horizontal lines denote a statistically significant difference ($P < 0.05$) based on a two-sided *t*-test. *$P < 0.05$, **$P < 0.01$, and ***$P < 0.001$ (Supplementary Fig. 17). **h** Inferred interaction networks based on population-level maximum growth rates in spatially separated MISTiC experiments. The size of the node represents the maximum growth rate in the 25 μm ($D_{25}$) condition and is computed as $(10 \ln D_{25}^{-1})^2$. The edge widths represent the interaction strength defined as $D_d = 1 - D_{25}D_{250}^{-1}$ where $D_{250}$ denotes the maximum growth rate in the 250 μm condition. Source data are provided as a Source Data file.

rescued (Fig. 5c, f, g). Notably, the reverse trend was observed for Δ*pheA*, wherein M was detected only in the highest supplemented F condition. In sum, the differential release profiles of M and F as a function of each producer strain's growth rate provides insight into the topologies of the inferred interaction networks (Fig. 5h).

Our results demonstrated that Δ*metA* and Δ*pheA* auxotroph strains have differential and context-dependent release rates of M and F, highlighting the environmental sensitivity of the system. To integrate our findings into a quantitative framework, we developed a computational model to represent M and F biosynthesis, uptake, diffusion, and amino acid dependent growth rates of Δ*metA* and Δ*pheA* (Supplementary Methods). Consistent with our data, we assume that (1) growth is limited by the concentration of the amino acid that the auxotroph is deficient in producing, (2) release of F by Δ*metA* is inversely proportional to its growth rate, (3) release of M by Δ*pheA* is proportional to its growth rate, and (4) the basal growth rate Δ*pheA* is larger than Δ*metA* which is attributed to differences in the metabolic consequences of each mutation (Fig. 5d). The model was fit to the population-level growth rates in the physically separated experiments using a genetic algorithm (Methods) and was able to recapitulate the trends across a range of conditions (Fig. 5b, c, f), demonstrating that the model's core assumptions were congruous with the data.

We next explored how different amino acid concentration influenced the inferred interaction network within MISTiC (Experiments 12–14, Table 1). The distance-dependent growth change of Δ*pheA* decreased with amino acid availability (Supplementary Fig. 15b, d, f, h), whereas this pattern was not evident for Δ*metA* (Supplementary Fig. 15a, c, e, g). In sum, these data demonstrated that the growth benefit provided by Δ*metA* was eliminated in the presence of trace amino acids.

In sum, the stability of the amino-acid auxotroph consortium was shaped by key variables including spatial arrangements, population size, amino acid availability, and growth rate-dependent amino acid release rates. The variation in each strain's growth across defined distances indicates whether metabolites mediating microbial interactions are in a saturated (e.g., below a concentration threshold to impact growth or not limiting for growth) or linear regime (e.g., limiting to growth). In addition, MISTiC enabled characterization of dynamic features of growth responses including biphasic growth (Supplementary Fig. 14b, c) and delays in the timing of maximum growth (Fig. 5c, inset).

## Discussion

We developed a microfluidic platform that integrates micrometer-level spatial patterning, temporal control of

environmental stimuli, and single-cell quantification of growth and gene expression within microbial consortia. In MISTiC, the distance between strains impacts local concentrations of diffusible molecules, which in turn dictate biological responses including growth rate, metabolic activity or cell state decision-making. Controlling the spatial arrangements of members of a consortium provides a quantitative mapping of the net environmental impact of a given strain on the dynamics of growth and gene expression in a partner strain. The dynamic phenotypic response of each strain illuminates the degree of sensitivity (linear regime) or decoupling (saturated regime) of the interaction, which would not be possible to study in real-time using standard culturing techniques. These quantitative features of microbial interactions are critical to understanding and engineering multi-species community stability and diversity[49,50].

We showed that distance can improve the reliability of information transmission in response to rapidly fluctuating signals. Consistent with this result, theoretical work has shown that spatial averaging by diffusion can improve the precision of gene expression by reducing noise stemming from transcriptional bursting[51]. In addition, stochastic modeling predicts that diffusion of a quorum-sensing chemical signal could reduce gene expression noise[39]. Our results suggest that spatial positioning can tune the trade-off between the fidelity of information transmission and the magnitude of the output response without imposing additional energetic costs to the cell[52]. Thus, distance between populations could be exploited as a design feature in microbial community engineering.

Changing the degree of spatial separation between the activator and repressor strains in the distributed gene circuit oscillator consortium yielded different outcomes in the oscillatory behaviors of each strain. The repressor oscillations abruptly vanished in the 250 μm interaction channel length, whereas the oscillations persisted in the activator strain in this condition. These data suggest that positive feedback enhanced the robustness of the oscillations across spatial separation, consistent with the critical role of positive feedback in expanding the spatial range of temporally synchronized gene expression patterns[53]. However, the negative feedback loop destabilized the temporal coordination of gene expression dynamics with distance, contrary to the stabilizing role of negative feedback observed in other systems[54].

In environments lacking M and F, the strength of the inter-strain auxotroph interactions depends on the rates of amino acid uptake, release and diffusion for constant population sizes. The opposing trend in the release profiles of M and F with increasing growth rate of Δ*pheA* and Δ*metA* (Fig. 5g), respectively, could

lead to feedback loops that destabilize the community under certain conditions. In *E. coli*, F is used for either protein synthesis or transported between the periplasm and cytosol[55], whereas M acts as a central hub methyl donor intersecting many pathways[55], potentially contributing to differences in release rates. Auxotrophic cross-feeding has been proposed as a strategy to enhance coexistence and stability among members of a consortium[46,56,57]. Our results suggest that strain co-existence and community stability depends on a critical population size, amino acid availability, and spatial structure[1,2,7], which limits the generalizability of amino-acid cross-feeding in real-world contexts.

MISTiC enables quantification of the effect of micrometer-level spatial separation on diffusion-mediated microbial interactions, in the absence of convection, transport, or cell-to-cell physical contact, by monitoring growth, gene expression, cell size, morphology, or emergence of cellular states as outputs. Although stable coexistence of community members can be difficult to achieve in batch or continuous culture[58], spatial separation of growth chambers within MISTiC maintains both strains for extended periods of time to study microbial interactions and community properties. In addition, the temporal variation of key diffusible compounds in microenvironments within MISTiC could be monitored via biosensors[59]. The media flow rate could be manipulated to study how the rate of diffusional loss shapes microbial interactions. To determine whether interactions stem from physical contact or diffusible compounds, future versions of MISTiC will contain a chamber to study the mixed community in addition to the spatially separated conditions.

There are limited techniques to investigate small bacterial populations ($\sim 10^2$ cells), which are prevalent in natural environments and can play important roles in human disease[60]. The uniform dimensions of the growth chambers dictate each strain's environmental impact and thus community functions. To interrogate the contribution of population size to microbial interactions and community stability, the growth chamber dimensions could be varied. To investigate multi-member consortia and higher-order interactions, the device could be modified for three or more interacting populations. In sum, this experimental platform could be adapted to study a diverse repertoire of organisms and mechanisms of diffusion-mediated interactions over multiple length-scales and increasingly complex spatial landscapes. A detailed understanding of how defined spatial arrangements influences community behaviors and interaction networks will advance our understanding of the role of spatial organization of microbiomes inhabiting diverse natural environments.

## Methods

**Microfluidic device fabrication**. A three-layer device was designed in AutoCAD that consisted of interaction channels, growth chambers, and main channels. The microfluidic master was pattered in three stages of photolithography using a micropattern generator (Heidelberg Instruments μPG 101). Unlike centrifuges, the spin coater protocol used for microfabrication is specified by an rpm. For the first layer, the silicon wafer was baked for 10 min at 200 °C and spin-coated at 4000 r.p.m. using SU-8 2000.5 (MicroChem) to generate 0.5 μm height. This layer was exposed to the interaction channels at 58 mW with a 47% dwell time using a 4 mm writehead, followed by a post-exposure bake for 30 min at 95 °C. The second layer was spin-coated at 3000 r.p.m. using a 26 : 1 mixture of SU-8 2000.5 to SU-8 3005, to produce 1.5 μm height. After aligning to the first layer, the wafer was exposed to the second patterning layer (growth chambers). Following an additional post-exposure bake, a third layer of SU-8 3025 photoresist was spin-coated at 3000 r.p.m. to generate 25 μm height. The wafer was exposed to the final layer consisting of the main channels, resistors, and inlets. Following a final post-exposure bake, the features were developed using SU-8 developer (MicroChem). The master was treated overnight with vapor phase (tridecafluoro-1,1,2,2-tetrahydrooctyl) trichlorosilane (Gelest) at room temperature. To fabricate each device, a 7 : 1 mixture of polydimethylsiloxane (Sylgard 184) to curing agent (Sylgard 184) was used to coat the master. After curing overnight at 100 °C, inlet and outlet holes were punched using a biopsy corer (WellTech). The surfaces were exposed to air plasma (Harrick Plasma PCD-32G) for 23 s to ionize the surface of the device to bond to the glass coverslips (ThermoFisher). Finally, the surfaces were bonded and baked

for 1 hr at 100 °C to seal the device channels. For each experiment, the microfluidic device was flushed with 0.5% Tween 20 (Sigma-Aldrich) to prevent cells from adhering to the device. To load the cells into the growth chambers, a vacuum pressure of 330 mm Hg was applied.

**Dye gradient experiment**. The chemical gradients in the interaction channels were analyzed by administering 10 μM fluorescein (Sigma-Aldrich) and water at a flow rate of 200 μL/h into individual main channels. Paired growth chambers ($n = 3$) connected by each interaction channel length were continuously imaged using a 600 ms exposure time. Fluorescence and phase-contrast Images were collected using a Ti-E Eclipse inverted microscope (Nikon) using the GFP filter (Chroma) 470 nm/40 nm (ex), 525/50 nm (em). To analyze the images, the fluorescence of each growth chamber and 1 μm increments along the length of each interaction channel at steady state were determined.

**Sender–receiver quorum-sensing experiments**. Sender and receiver plasmids (Supplementary Fig. 2) were constructed using standard Gibson assembly protocols using primers synthesized by Integrated DNA Technologies and verified by Sanger Sequencing (Functional Biosciences). The sender (A6c_LuxI_GFP[48]) and receiver (E2c_LuxR_RFP or pJH9-35) plasmids were transformed into *E. coli* strains BW27783[61] and MG1655Z1[62] (Table 2), respectively. An initial set of cultures were inoculated into LB media (Lennox, Sigma-Aldrich) containing 25 μg/mL chloramphenicol (Sigma-Aldrich) and cultured overnight at 37 °C with shaking. After ~16 h, 1 μL of the cultures were diluted into 3 mL LB media containing 25 μg/mL chloramphenicol and incubated at 37 °C with shaking to early stationary phase (OD600 0.7–1.1). Next, we measured the OD600 of these cultures and centrifuged 1 mL at $3500 \times g$. The supernatant was removed and the pellet was resuspended to a final OD600 of ~20. Cells were loaded into the device such that each growth chamber had two to three cells at the beginning of the experiment. In each experiment, the device was connected to three syringes (5 mL) containing LB media supplemented with 25 μg/mL chloramphenicol, 0.1% Tween 20 (Sigma-Aldrich), and 62.5 ng/mL anhydrotetracycline hydrochloride (Cayman Chemicals), as well as a fourth syringe (5 mL) containing the same media supplemented with 0.1% arabinose (Sigma-Aldrich). During the microscopy experiment, the microfluidic device was incubated at 37 °C in a custom-designed temperature incubation chamber. The main channels were flushed at a rate of 300 μL/h to wash away excess cells from the growth chamber. The flow rate of the inlet containing arabinose ($I_{22}$, Fig. 1) and the corresponding inlet on the opposite side ($I_{11}$) were set to 10 μL/h to prevent cell growth and clogging within the inlet and resistor and to reduce pressure differences across the device. Fluorescence and phase-contrast images were collected using a Ti-E Eclipse inverted microscope (Nikon) every 7 min at 21 different positions. Fluorescence was imaged using the following filters (Chroma): GFP: 470 nm/40 nm (ex), 525/50 nm (em) or RFP: 560 nm/40 nm (ex), 630/70 nm (em). The device was incubated for a period of time to allow cells to grow and divide. After the growth chambers had filled with cells, the media was switched to test conditions described in Table 1. For Experiment 1 (Table 1), the arabinose inlet ($I_{22}$, Fig. 1) and the corresponding inlet on the opposite side ($I_{11}$, Fig. 1) were switched to 200 μL/h and the flow rate through the remaining inlets ($I_{12}$, $I_{21}$) were set to 0 μL/h. The forced oscillation experiments (Experiments 3 and 4, Table 1) used 10 mL syringes to extend the duration of the experiment. Flow rates of the 0.1% arabinose (inlet $I_{22}$) and 0% arabinose (inlet $I_{21}$) were alternated out of phase between 200 and 0 μL/h for a period of time. One of the receiver inlets ($I_{11}$) flowed continuously at a rate of 200 μL/h and the other inlet ($I_{12}$) was set to 0 μL/h for the duration of the experiment.

**Dual-feedback oscillator experiments**. The *E. coli* strain CY027 was transformed separately with plasmids pC220 and pC239 or pC236 and pC239 to construct the activator and repressor[43], respectively using a standard chemical transformation protocol (Table 2). Overnight cultures were inoculated into LB media (Lennox) containing 50 μg/mL kanamycin and 100 μg/mL spectinomycin and incubated overnight at 37 °C with shaking. After ~16 h, 1 μL of the overnight cultures were diluted into 3 mL LB media and incubated at 37 °C with shaking to early stationary phase (OD600 0.7–1.1).

Cells were loaded into the device following the procedure specified above. Following cell loading, the microfluidic chip was placed in the custom-designed temperature incubation chamber at 37 °C. All four inlets were connected to syringes (10 mL) containing LB media with kanamycin (50 μg/mL), spectinomycin (100 μg/mL), and 0.1% Tween 20. Syringes connected to inlets $I_{22}$ and $I_{11}$ also contained 1 mM isopropyl β-D-1-thiogalactopyranoside (IPTG) (Sigma).

The cells were initially grown in the device at 37 °C with inlets $I_{12}$ and $I_{21}$ flowing at 200 μL/h, and inlets $I_{22}$ and $I_{11}$ flowing at 10 μL/h to prevent cell growth and clogging. Phase-contrast and fluorescence images were collected every 7 min at 21 different positions. Once the growth chambers were filled with cells (Table 1), the inlets ($I_{12}$ and $I_{21}$) containing the pre-culture media were set to 0 μL/h and the inlets ($I_{11}$ and $I_{22}$) containing the test media were set to 200 μL/h.

**Amino acid auxotroph experiments**. *E. coli* strains Δ*metA*[63] and Δ*pheA*[63] (Table 2) were transformed with plasmids A6c_GFP[64] and A6c_RFP[64], respectively, using a standard chemical transformation protocol. The plasmids harbored

**Table 1 MISTiC experimental conditions.**

| Experiment | Descriptor | Media switch (min) | Pre-culture condition | Test condition | Outliers |
|---|---|---|---|---|---|
| 1 | Q.S. (step response) | 105 | aTc | aTc + ara | 1;1;0;0 |
| 2 | Q.S. (step response, inverted) | 240 | aTc | aTc + ara | 4;0;1;2 |
| 3 | Q.S. (forced oscillator, 2 hr) | 140 | aTc | aTc ± ara | 1;1;1;5 |
| 4 | Q.S. (forced oscillator, 1 hr) | 184 | aTc | aTc ± ara | 0;0;1;3 |
| 5 | Distributed gene circuit oscillator | 217 | NA | IPTG | 0;0;3;1 |
| 6 | Auxotroph control | 218 | 0.25× AA + IPTG | 0.25× AA | 0;0;1;2 |
| 7 | Auxotroph (−F, −M) | 420 | 1× AA* + 0.1× F/M + IPTG | 1× AA* | 3;2;0;1 |
| 8 | Auxotroph (mixed) | 60 | 1× AA* + 0.1× F/M + IPTG | 1× AA* + IPTG | 5;2;2;1+ |
| 9 | Auxotroph (ΔmetA control) | 110 | 1× AA* + 0.1× F/M + IPTG | 1× AA* + IPTG | 1;2;3;8+ |
| 10 | Auxotroph (ΔpheA control) | 90 | 1× AA* + 0.1× F/M + IPTG | 1× AA* + IPTG | 3;2;2;7+ |
| 11 | Auxotroph (ΔmetA rescue) | 24 | 0.5× AA* + 1× M + 0.05× F + IPTG | 0.5× AA* + 1× M | 0;0;0;1 |
| 12 | Auxotroph (All AA) | 710 | 0.1× AA + IPTG | 0.003× AA | 1;0;0;0 |
| 13 | Auxotroph (All AA) | 148 | 0.1× AA + IPTG | 0.005× AA | 1;0;0;3 |
| 14 | Auxotroph (All AA) | 285 | 0.2× AA + IPTG | 0.02× AA | 1;2;0;0 |

Pre-culture conditions refer to the environment within the microfluidic device from the beginning of the experiment to the time of the first media switch. Test culture conditions refer to the media conditions following the first media switch. Quorum-sensing experiments were performed in Luria Broth (LB), whereas remaining experiments used MOPS EZ Rich Defined Medium, with the modifications specified above. AA refers to EZ amino acid solutions containing all amino acids and AA* refers to an amino acid solution lacking methionine (M) or phenylalanine (F). In experiments where F and M were added separately, 1× F and M refers to 0.4 mM and 0.2 mM, respectively. The pre-culture amino acid fraction was varied to control cell growth and fluorescent reporter expression. Outliers refer to the number of paired growth chambers that were excluded from the analysis out of n = 10 biological replicates for each interaction channel length (25 μm;50 μm;100 μm;250 μm) based on a set of specific criteria (Methods). +For these experiments, the numbers represent single outlier growth chambers that were excluded from the analysis out of n = 20 total biological replicates for 25 μm, 50 μm, 100 μm, and 250 μm interaction channels (Methods).

**Table 2 Strains used in study.**

| Strain identifier | Strain background | Plasmid(s) | Reference |
|---|---|---|---|
| Sender | BW27783 | A6c_LuxI_GFPa | [61] |
| Receiver | MG1655z1 | E2c_LuxR_RFPa | [62] |
| Activator | CY027 (E. coli ΔlacI ΔaraC ΔsdiA Ptrca-cinR Ptrca-rhlR) Addgene #72402 | pC220 (Addgene #65877) and pC239 (Addgene #65953) | [43] |
| Repressor | CY027 (E. coli ΔlacI ΔaraC ΔsdiA Ptrca-cinR Ptrca-rhlR) Addgene #72402 | pC236 (Addgene #65951) and pC239 (Addgene #65953) | [43] |
| ΔmetA | BW25113 | A6c_GFPa | [63] |
| ΔpheA | BW25113 | A6c_RFPa | [63] |

aThe symbol indicates plasmids that were constructed for this study.
All other constructs were derived from the indicated references.

an IPTG-inducible fluorescent reporter. An initial set of cultures were inoculated into LB media (Lennox) containing chloramphenicol (25 μg/mL) and incubated overnight at 37 °C with shaking. After ∼16 h, 1 μL of the overnight cultures were diluted into 3 mL of LB containing 25 μg/mL chloramphenicol and 1 mM IPTG (Sigma-Aldrich) and incubated at 37 °C with shaking until early stationary phase (OD600 0.7–1.1).

The cells were loaded into the device following the procedure outlined above. Following cell loading, the microfluidic chip was placed in the custom-designed temperature incubation chamber at 37 °C. The media always contained 1× MOPS Buffer (Teknova), 1× ACGU mix (Teknova), chloramphenicol, 0.1% Tween 20, 1.32 mM potassium phosphate dibasic (Teknova), and 0.2% glucose (Teknova), whereas the amino acid composition varied across experiments (Table 1). The amino acid solutions consisted of either EZ Amino Acids (AA, Teknova) or a modified amino acid solution (AA*) (Table 1). The AA* solution consisted of 0.4 mM L-asparagine (VWR), 0.01 mM calcium pantothenate (VWR), 0.2 mM L-histidine (VWR), 10 mM L-serine (VWR), 0.8 mM L-alanine (Fisher Scientific), 0.4 mM L-lysine (Fisher Scientific), 0.1 mM L-tryptophan (Fisher Scientific), 0.4 mM L-aspartic acid (Dot Scientific), 0.1 mM L-cysteine (Dot Scientific), 0.8 mM L-glycine (Dot Scientific), 0.4 mM L-isoleucine (Dot Scientific), 0.8 mM L-leucine (Dot Scientific), 0.01 mM para-amino benzoic acid (Dot Scientific), 0.4 mM L-proline (Dot Scientific), 0.4 mM L-threonine (Dot Scientific), 0.6 mM L-valine (Dot Scientific), 5.2 mM L-arginine (Sigma), 0.01 mM di-hydroxy benzoic acid (Sigma), 0.6 mM L-glutamic acid (Sigma), 0.01 mM para-hydroxy benzoic acid (Sigma), 0.01 mM thiamine (Sigma), 0.2 mM L-tyrosine (Sigma), and 0.6 mM L-glutamine (Acros Organics). In minimal media supplemented with AA*, varying concentrations of methionine (Dot Scientific) and/or phenylalanine (Dot Scientific) were added. The AA amino acid solution consisted of all components in AA* plus 0.2 mM methionine and 0.4 mM phenylalanine.

The cells were grown for a period of time at 37 °C with 1 mM IPTG prior to the media switch as described in Table 1 to fill the growth chambers. Phase-contrast and fluorescence images were collected every 10 min at 21 different positions. After the growth chambers were filled with cells, the inlets (I₁₂ and I₂₁) containing the pre-culture media were set to 0 μL/h and the inlets (I₁₁ and I₂₂) containing the test media were set to 200 μL/h.

**Amino-acid measurements.** The ΔmetA and ΔpheA strains were inoculated into LB (Lennox) containing chloramphenicol (25 μg/mL) and grown overnight at 37 °C with shaking. After 16 h, 10 μL of the cultures were transferred into 3 mL of fresh LB containing chloramphenicol (25 μg/mL) and incubated at 37 °C with shaking until early stationary phase (OD600 0.7–1.1). Immediately following, the cultures were centrifuged at 3500 × g for 5 min, supernatant was removed, and the cells were inoculated into MOPS EZ Rich Defined Medium lacking M and F at an initial OD600 of 0.05. For the ΔmetA strain, 0, 2, 5, 10, or 200 μM M was added to the media. For ΔpheA strain, 0, 4, 10, 20, or 400 μM F was added to the media. The cultures were incubated at 37 °C with shaking for 3 h. After recording the OD600 of each culture, the cells were centrifuged at 3500 × g for 10 min, the supernatant was filtered by a 0.2 μm filter (GE Healthcare) and the concentrations of M or F were measured with a fluorometric assay kit (BioVision) or by liquid chromatography-mass spectrometry (LC-MS), respectively. Concentrations of M in the filtered conditioned media of ΔpheA cultures were measured with a fluorometric methionine assay kit (BioVision) with a 0.5 μM limit of detection. Raw fluorescence measurements were converted to methionine concentrations using a standard curve.

The analysis of F concentrations in the filtered conditioned media of ΔmetA was performed on a Shimadzu LC-MS2020. All solvents and reagents used for analysis were HPLC grade or higher quality. Methanol and formic acid were sourced from Fisher Scientific and Acros Organics, respectively. Water was prepared in house with a Millipore Milli-Q water purification system. Separations were performed at 40 °C on a Discovery BIO wide pore C5-5 column (15 cm × 2.1 mm × 5 μm) from

Millipore-Sigma with a paired Supelguard (2 cm × 4 mm × 5 μm) guard column. The running buffer was a binary gradient of water with 0.1% v/v formic acid (Buffer A) and methanol (Buffer B) according to the following protocol: 4 min at 5% B, a linear gradient from 5% to 20% for 4 min, a linear gradient from 20% B to 95% B for 2 min, 2 min at 95% B, a linear gradient from 95% B to 5% B for 2 min, equilibration at 5% B for 6 min. The total flow rate was 0.2 ml min⁻¹. Under these conditions, methionine and phenylalanine eluted at 3.8 and 5.8 min, respectively. The ion source was operated in electrospray ionization mode with a cone voltage of 4.5 kV, the interface was held at 400 °C, and the desolvation line at 250 °C. The dry nitrogen was supplied to the nebulizer at 1.5 L/min and drying gas at 15 L/min. The mass spectrometer was run in selective ion monitoring mode for monitoring $m/z$ 150 for methionine and $m/z$ 166 for phenylalanine with a scan time of 1 s. Standards were prepared for each run by adding known concentrations of methionine and phenylalanine to fresh media. The standard curve was run before and after the sample batch and each sample was run twice for technical replicates.

**Auxotroph community batch culture experiment**. Separate culture tubes containing LB with chloramphenicol (25 μg/mL) were inoculated with $\Delta metA$ or $\Delta pheA$ and incubated overnight at 37 °C with shaking. After 16 h, the cultures were diluted in 5 mL of EZ Rich Medium (Teknova) containing chloramphenicol (25 μg/mL) and 1 mM IPTG and lacking M and F at a final OD600 of 0.05. The initial ratio of $\Delta metA$ to $\Delta pheA$ was 10:1, 1:1, or 1:10 ($n = 3$, for each starting ratio). The cultures were incubated at 37 °C with shaking for at least 24 h before transferring the community to fresh media using a 1:100 dilution. At this transfer time, the OD600 of each culture was measured and a 2 μL sample was spotted onto a glass slide for cell counting with microscopy (20× magnification) on a Nikon Eclipse Ti. Four images comprising four distinct fields of view were taken of each sample and each image was a composition of phase contrast, GFP and RFP channels. Subsequently, ImageJ was used to extract the number of $\Delta metA$ cells from the GFP channel and $\Delta pheA$ cells from the RFP channel for each image.

**Population-level image analysis**. For Experiments 1, 3, 6, 7, and 11–14 (Table 1), individual growth chambers were segmented in DeepCell[65]. Five neural networks were trained on 21 randomly selected images and binary masks (made using FIJI image analysis software[66]), which specified the growth chamber positions. The trained model was used to analyze the remaining microscopy images. The results of the trained networks (two to five depending on segmentation accuracy) were averaged to improve segmentation accuracy.

For Experiments 2, 4, and 5 (Table 1), growth chambers were segmented using custom code in Python that aligned each growth chamber across all time points. A binary mask denoting the growth chambers was applied to all time points. The DeepCell and alignment methods generated nearly identical fluoresence time-series data.

In each analyzed image, custom code (Python) was used to label the binary mask with the growth chamber positions and total areas and compute the average fluorescence intensity of each growth chamber. Segmented regions less than or greater than 1000 and 3500 pixel area were eliminated from the data set. Specific criteria were used to eliminate outliers from the datasets including (1) infrequent pressure fluctuations leading to loss of cells from the growth chambers, (2) device bonding issues leading to collapsed interaction channels or cells that enter the interaction channels, (3) growth chambers with unoccupied regions, (4) abnormal cell growth that significantly altered the total number of cells in the growth chamber, or (5) cell growth in the main channels that may have generated different media diffusion rates into the growth chambers. In all physically separated experiments (Experiments 1–7 and 11–14, Table 1), the connected chamber was excluded from the data set if a growth chamber was identified as an outlier based on these criteria (Table 1).

**Population-level fluorescence time-series analysis**. Fluorescent time-series measurements for each growth chamber in Experiments 6, 7, and 11–14 (Table 1) were analyzed by bootstrapping. Using this method, the biological replicate curves for a given interaction channel length were randomly sampled 10,000 times with replacement. In Experiment 1 (Table 1), background fluorescence was subtracted from the data by subtracting the minimum RFP fluorescence intensity across all growth chambers for model fitting (Fig. 1c, d).

The P-values for all bootstrapped datasets were computed by bootstrap hypothesis testing. Here, the null hypothesis $H_0$ assumes that a sample of size $n$ with mean $x^*_{obs}$ and a sample of size $m$ with mean $y^*_{obs}$ are derived from the same population. This test is performed as follows:

Calculate the sample mean difference, $t^*_{obs}$, as $t^*_{obs} = x^*_{obs} - y^*_{obs}$

Merge two samples into one set of $n + m$ observations.

Draw a bootstrap sample of $n + m$ observations with replacement from the merged set.

Calculate the mean of the first $n$ observations, $x^*$ and compute the mean $y^*$ for the remaining $m$ observations in the bootstrap sample. The test statistic $t^*$ is evaluated as $t^* = x^* - y^*$.

Repeat steps 3 and 4 $B$ times where $B \geq 1000$.

Evaluate the P-value as: P-value = number of times where $t^* > t^*_{obs}$ divided by $B$.

Reject $H_0$ if P-value $\geq \alpha$, where $\alpha = 0.05$.

In the forced oscillation experiments (Experiments 3 and 4, Table 1), a peak finding algorithm (Python) was applied to the time-series gene expression data at steady state with minimum inter-peak threshold of 21 min. The amplitude was computed by subtracting the minimum and maximum of each oscillation and dividing this value by two. To calculate the SNR, a moving mean computed over 20 points was subtracted from the data. The power spectra for each replicate was calculated using Welch's method (Python) with a Hamming window applied across the length of the time-series. The power spectra were filtered to exclude frequencies lower than the signal bandwidth. The signal was defined as the total power of the signal bandwidth. The noise was computed as the total power of frequencies larger than the signal bandwidth. The power spectra for all the replicates for a given interaction channel length were randomly sampled with replacement 10,000 times. For each iteration, the SNR ratio was computed by dividing the signal by the noise. Bootstrap hypothesis testing, as described above, was used to compute P-values.

For the distributed gene circuit oscillator experiment (Experiment 5, Table 1), the fluorescence intensity of each fluorescent reporter was normalized by subtracting the global minimum of the reporter across all replicates, dividing by the global maximum of fluorescence across all replicates, and applying a moving mean of 20 time points to the data. A peak finding algorithm (Python) was applied to detect peaks with a minimum inter-peak distance of 70 min and a minimum peak height of 0.015 by analyzing the data after the media switch. The number of peaks detected, the amplitude of expression at each peak, and the distance between subsequent peaks were computed for each replicate.

In the spatially separated auxotroph experiments (Experiments 6, 7, 11–14, Table 1), the fluorescence background for each reporter was subtracted from the data and then the time-series was normalized by dividing by the maximum value. The change in fluorescence per unit time ($\Delta F\, \Delta t^{-1}$) was computed by determining the slope of a line fit to a 10 time point moving window and then multiplying by negative one. The global maximum of $\Delta F\, \Delta t^{-1}$ corresponded to the maximum growth rate. The doubling time was calculated using times the following equation:

$$\text{Doubling time} = \frac{\ln(2)}{\max(\Delta F \Delta t^{-1})}.$$

In Experiment 11 (Table 1), the $\Delta F\, \Delta t^{-1}$ curves displayed a biphasic trend. To characterize the growth rate at each peak, the $\Delta F\, \Delta t^{-1}$ time-series was analyzed between the time point of the media switch and the time point corresponding to 25% of the maximum fluorescence. The local maxima within this time window were identified using the findpeaks algorithm (Python). The bootstrapped $\Delta F\, \Delta t^{-1}$ time-series were aligned by the first peak and the doubling times at the global maximum were calculated as described above. For the second growth phase, the doubling time was calculated at the maximum $\Delta F\, \Delta t^{-1}$ for the period of time between the global maximum and the time point corresponding to 25% of the maximum fluorescence.

**Single-cell image analysis**. Single-cell metrics were obtained with a custom machine learning approach implemented in Python with the Keras API running on top of TensorFlow[67]. We used two convolutional neural networks with U-Net architecture. First, we performed segmentation of individual cells in each image and then tracked each of the segmented cell instances over time. The segmentation network takes as an input the phase contrast images of cells grown in MISTiC and for each image and yields a binary mask segmenting the cells from the background. Training data were obtained from a separate experiment imaging fluorescently labeled $E.\ coli$ at 60× magnification with phase-contrast and fluorescence images collected every 10 min. We used the fluorescence images to generate binary segmentation masks of the cells, which then served as the ground truth for the phase contrast images used for network training. A total of 1066 images were curated this way. The network was trained for 100 epochs using a stochastic gradient descent optimizer and a pixelwise weighted loss function to enforce the learning of narrow borders between adjacent cells. To minimize overfitting of the network to the training data, random affine transformations and elastic deformations were applied in real-time during the training process.

Cell tracking was performed with a separate U-Net similar to a method reported previously[68]. The input for this network is a set of consecutive binary segmentation masks. For each cell in the current segmentation, the network predicts the cell in the previous segmentation image from which the current cell was derived. This backwards tracking approach eliminates the need for the network to learn occurrences of cells leaving the chamber and reduces the number of classes to two (the tracked cell and the background). Using segmentations from the mixed auxotroph experiment, we curated 2656 sets of training images with a custom script in MATLAB. Training occurred for 200 epochs using an Adam optimizer and a class-weighted categorical cross-entropy loss function. Similarly, data augmentation was performed to reduce overfitting of the data.

Following segmentation and tracking, the raw output was processed with custom code in Python to reconstruct cell lineage and obtain single-cell metrics. The instantaneous growth rate of each cell was computed from the cross-sectional area recorded during the 100 min window (10 data points) immediately following that instant. Growth rate was computed by fitting a line to each 100 min window and then dividing the slope of the line by the average cell area during that time interval. For all analyses, a minimum tracking duration of 100 min was imposed to enforce consistent computation of growth rate. For all analyses involving growth

rate, statistical outliers were identified using a modified $z$-score computed on the chamber averaged growth rates at each time point

$$M_i = \frac{0.6745(x_i - \tilde{x})}{M_d},$$

where $\tilde{x}$ represents the median growth rate and $M_d$ denotes the median absolute deviation[69]. Statistical outliers were detected using a threshold of $M_i > 3.5$. Growth chambers with more than one time point registering as an outlier were excluded from the analysis (Table 1). Experimental outliers occurred primarily due to segmentation and tracking errors caused by loss of focus or empty chambers at specific positions. Outliers were considered separately for each strain.

**Model fitting.** Custom code (MATLAB) was used for computational modeling. An ordinary differential equation model was developed to study inter-strain communication via chemical signal diffusion (quorum-sensing). Detailed descriptions of the diffusion and gene expression models are in the Supplementary Methods. The general mathematical form of the equations describing the concentration of AHL or fluorescein in each discretized spatial region is

$$\dot{x}_i = D(x_{i-1} + x_{i+1} - 2x_i) - \gamma x_i,$$

where $x_i$ and $x_{i+1}$ represent concentrations in adjacent regions of the device. The parameters $D$ and $\gamma$ denote the diffusion and degradation rates of the diffusible molecule, respectively. For the gene expression model, the general mathematical form for modeling transcription is

$$\dot{B}_m = \propto_B \frac{A^n}{K^n + A^n} - \gamma B_m,$$

where $A$ and $B_m$ represent a transcription factor and its regulated transcript, respectively. The parameters $\propto_B, n, K,$ and $\gamma$ denote the maximum transcription rate, Hill coefficient, half-maximum concentration or binding affinity and mRNA degradation rate, respectively. The general mathematical form for representing time delays due to sequential protein assembly, fluorescent protein maturation or media switching is

$$y_j = a(y_{j-1} - y_j) \, for \, j = 1 : N.$$

The species $y_N$ represents the time-delayed species $y_1$ and the delay time is computed by $N \cdot a^{-1}$.

The model was simulated using ode23s (MATLAB). A model with a variable number of delay equations was fit to the data using a genetic algorithm. The algorithm identified a best estimate for the parameter values and an optimal model structure by adjusting the number of delay equations to minimize the $L^2$-norm between the model and the data. First, 100 parameter sets were randomly sampled using an upper and lower bound for each parameter. For each parameter set, the model was simulated and the $L^2$-norm between the model and the data was computed. The parameters were ranked from lowest to highest $L^2$-norm. The first parameter set (lowest $L^2$-norm) was averaged with parameter sets 2-10, generating 9 new parameter sets. These parameter sets were combined with 81 randomly sampled parameter sets using an upper and lower bound for each parameter. This procedure was repeated until the $L^2$-norm did not change significantly with additional iterations. The best estimates for the parameters are listed in Supplementary Table 3.

The parameters of the amino-acid cross-feeding model (Supplementary Methods) were fit using a genetic algorithm. The genetic algorithm can be most efficient with high-order systems and many unknowns. One of the challenges with the genetic algorithm is there is no proof of convergence and the rate of convergence can be slow if the initial guesses on the parameters are far from the minimizing set and the bounds on the parameters are too broad. In order to overcome these challenges, careful consideration was taken to determine the lower and upper bounds for each parameter. Initially, bounds were determined based on biologically relevant and feasible values. In addition, experimental observations were used to infer necessary relationships between parameters. The bounds on the parameters were adjusted accordingly. After this, the genetic algorithm was executed until the error became invariant for a sequence of 10 generations. As the genetic algorithm is not optimal, it is possible to arrive at slightly different values if we were to run the genetic algorithm longer or reinitiate at new random initial conditions. However, the qualitative fits remain fairly close, as do the parameter values. Nevertheless, given experimental error, it is not in our benefit to achieve an optimal fit, since such a fit does not imply better prediction of quantitative values of parameters.

**Reporting summary.** Further information on research design is available in the Nature Research Reporting Summary linked to this article.

## Code availability
The code used for computational modeling is available at 10.5281/zenodo.3748013.

## Data availability
The source data for Figs. 1–5 are available as Source Data files. The raw data files are available at 10.5281/zenodo.3748013. All other relevant data are available from the authors upon reasonable request.

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

## Acknowledgements

We thank John Crooks and Jeremy Schroeder for assistance with the laser pattern generator, and Michael Chevalier, Ryan Clark, Bhuvana Krishnaswamy, Freeman Lan, Austin McDaniel, Urbashi Mitra, and Nimish Pujara for helpful discussions. Research was sponsored by the Army Research Office and was accomplished under Grant Number W911NF-19-1-0269 and W911NF-17-1-0296 and the National Institutes of Health Grant Number R35GM124774. S.G. was supported by NIH/NIGMS under award number T32 GM008293.

## Author contributions

O.S.V., P.A.R., and S.G. designed the research. S.G. and T.D.R. carried out experiments. S.G., J.L.G., and O.S.V. performed computational modeling. O.S.V., P.A.R., S.G., and T.D.R. discussed data analyses and S.G., O.S.V., and T.D.R. performed the analyses. O.S.V., S.G., P.A.R., and T.D.R. wrote the manuscript. J.L.G. and T.D.R. carried out the metabolite measurements. O.S.V. secured funding.

## Competing interests

The authors declare no competing interests.
