## [Peer Review File · Nature Communications]

Reviewers' Comments:

Reviewer #1:

Remarks to the Author:

The authors have developed a microfluidic system that enables elucidation of molecular communication between cells in controlled microenvironments. The fluidic system provides for signal molecule transport across defined distances bounded by cells with different genetic backgrounds. The channels are constructed with a 0.5 micron height, which should severely restrict cell transport. The cells under investigation are engineered to (i) express fluorescent marker proteins in response to molecular stimuli (arabinose and 3-oxo-C6-AHL), or (ii) grow differentially in response to exogenously added amino acids. In addition, for the former, the authors have developed a relatively simple mathematical model that describes the expression of fluorescent markers within the engineered cells as a function of time. They have referred to the complete system as "MISTiC" for Mapping Interactions across Space and Time in Communities. In this way, the combination of the model and the microfluidic device enables discrimination of the signal transduction processes known to affect cell phenotype, which are particularly important for understanding behaviors within microbial consortia. Also, they have used the AHL signaling mechanism of *Vibrio fischeri*, which has essentially no exogenous impact on *E. coli*, the model system of choice.

The manuscript is expertly prepared and the experiments appear to have been performed with care, including mathematical, statistical, and graphic representation of the data. The synthesis of these methodologies represents a fine contribution to the field in that there are few reports that provide the systematic amalgamation of the potentially disparate components that comprise this report. That said, there are several reports from several groups that have performed similar studies that have not been recognized in the current manuscript. The accumulation of these reports and their direct relevance to the current report, point to the question of novelty and partially diminish this reviewer's enthusiasm for this manuscript as a contribution in *Nature*.

For example, there are many groups who have examined cell signaling, including bacterial cell-cell signaling and even quorum sensing, in carefully controlled geometries. Atencia et al. (*Lab Chip*, 2009) created a microfluidic palette that is used to create carefully controlled chemical gradients in the absence of convective flows; they show how their system can be used to interrogate chemotaxis in response to carefully controlled glucose gradients. Jayaraman and coworkers (*Lab Chip*, 2012; *J. Bact.*, 2014; *Biomed Microdevices*, 2014) have published an array of devices that enable carefully controlled microenvironments for the study of biological phenotype, including responses to chemical gradients. Diao, et al. (*Lab Chip*, 2006) constructed a system that interrogates bacterial responses that overlap with quorum sensing. Bentley and coworkers (*Biomicrofluidics*, 2017; *Biotechniques*, 2017, *Lab Chip*, 2015; *Biomaterials*, 2012) developed methodologies that demonstrate quorum sensing (QS) interrogation and control in similar systems. Correspondingly, they have developed agent based models to interrogate QS behaviors as a function of the molecular mechanisms of QS signal transduction (*PLoS Comp Biol*, 2016). These are a few of the many examples. Some of these methodologies reveal single cell behaviors, some reveal expression patterns at nearly single cell level, all provide visual basis for interpreting the phenotype in response to carefully controlled gradients. None of these reports, however, put all of these together in a way shown in the current manuscript.

There were a few issues, however, that I feel should be addressed in addition to the issue of novelty. The first has to do with the most basic premise of the paper. The current manuscript emphasizes the distance between cells, yet recognizes that it is the concentration of the signal molecules that determine the fate of the biological response. Fixing this issue is relatively simple, but as currently prepared, the manuscript promulgates a suggestion that distances matter. This is only the case when convection, diffusion, cell migration, signal generation, and signal uptake are all controlled and static. Another relatively simple issue has to do with the use of diffusion rate and the systematic absence of diffusivity and a Fickian diffusion coefficient. The data could be recast as

a function of the expected results given the diffusion coefficient and quiescent flow. The studies on fluorescein for example, could be used to abstract a diffusion coefficient, that in turn, could be used to validate that there is no convective flow in the device. Without this, there are limited control experiments that provide this insight. How do we know there was no convection? Pressure swings in these devices are common and difficult to control. Along these lines, were studies done that demonstrate the absence of cell migration or motility? For example, if sterile media were provided on one side and cells and growth media on the other, would the sterile side stay sterile for the duration of the experiments here? A third issue might be a result of reviewer misunderstanding. There are many experiments wherein the fluorescence is observed to decrease, and rapidly. Indeed, the degradation rate constants in the model for LasI and GFP are non-zero (and different). Some clarification is needed relative to the rapid loss of fluorescence observed in the square wave experiments (the loss seems to be nearly the same rate as the increase), while the simple "ON" experiments seem to have a very stable fluorescence. Isn't it well known that GFP is stable and less well known that LasI may be less stable? Independent reference relative to stability would help. The data suggest that these two proteins have significant turnover rates, which is not expected and not shown in control studies. Perhaps either the number of cells in the co-culture experiments might be changing in time (owing to dilution of the fluorophore in the presence of cell growth; there were no data on the numbers of sender or receiver cells) or that the degradation of the fluorophores is dramatically important and affected by undescribed factors. Please clarify these dynamics. Finally, the auxotroph experiments seem to be of limited utility. Cells receive many signals from a secretome, they grow, and they otherwise respond. The effects of the mutations are not simply reversed by the addition of the amino acid. The metabolic consequences of these mutations are widespread within the given host (for example, tyrosine synthesis also needs pheA). The fundamental insight obtained by these experiments is not clear.

There are a few relatively minor issues:

1. P. 5. What is D_2 ? Is there a way to estimate this?
2. P. 5. How many cells are there? 150-200? I thought the phase contrast images could provide this number. It is supposed to be constant in the first sets of experiments.
3. P. 3 (chemical signaling, multispecies) & P. 14 (metabolic burden). Should be careful to cite original references.
4. S1. Why no cell number equations? The mutation experiments aren't modeled...reinforcing their limited value.
5. S3. It seems like degradation tags would have been appropriate for the oscillatory input experiments.
6. Fig S6. When one looks at these data and the overlaid gray bars, it is hard to directly visualize a response to the chemical input. This is especially problematic at low times (<400 min) and longer times (>500 min). This was noted in the manuscript, but it seems problematic in this figure.
7. Fig S8. What happens at 300 min?
8. Fig S10. Does the data in "a" make sense?
9. Provide the host cells and their origins or all experiments. I believe the knockouts are isogenic to the host.

Reviewer #2:

Remarks to the Author:

The manuscript presents a microfluidic device called MiSTiC that can be used to study how distance and diffusion affect the exchange of molecules between two bacteria populations. They tested the device with AHL as a molecule of exchange and with two population of auxotrophs that require each other to survive.

I have several questions/comments I would like the authors to respond to.

- MiSTiC is referred as a tool to study spatially structured environments, however, it seems that the only variable that can be studied is the distance of the channel, what basically varies the diffusion of the molecules from one chamber to the other. For example, if more AHL reaches the second chamber more fluorescence is found, something rather logic and expected. The same can be found in the exchange of amino acids. It is well known that diffusion is key in bacterial communities and these experiments verify that.

-in the abstract, the authors state "Our findings suggest generalizable principles that could be exploited to program the spatiotemporal behaviors of microbial consortia." However, it is not clear what are these principles and how they can be used, and importantly how they are new or unknown before.

-I find the introduction rather short and lacking more description of previous microfluidic approaches to study communities.

-the authors should study how changes in the flow rate (input media flow) affect diffusion and exchange in the device.

-line80. The characterization with the fluorescent dye is not really well studied. What is the diffusion coefficient of that dye? why the variation of the fluorescence with the distance is not symmetric? Is the dye also degraded on time?

-In relation to the comment above, why the device is not designed in a symmetrical manner? In figure 1 a, the flow in blue could go in the opposite direction creating a symmetric display. If that is changed, would this affect to the symmetry of the fluorescent dye?

-it is also unclear what is the metabolic state of the cells in the chamber, what are the cells that are washed out? The daughter cells? Are you concentrating older cells in the chamber? Do they replicate in the same manner across the length of the chamber?

-130-131. This is basically due to the fact that less AHL is reaching the cells, something not very surprising as when you add less AHL to cells in other configurations (tubes, flasks, plate..) you also get a lower response. Also there you mention the response time as not altered, but this is depending on the time scale you consider and the limitations of the measuring times.

-the model results seem rather expected and logical. How do the authors believe the model can be used to generate new insights in the system?

-198-200. Maybe trying other QS molecules can help to prove this point.

-it seems that the noise in RFP expression could come from the specific times of expression, folding and degradation. Are these parameters known for the specific RFP used in this study? Maybe you can give values and learn the minimum time required to obtain a good response.

-289, would not be possible to quantify the cells being washed out? Or leaving the chamber? That might be more precise. Fluorescence decay can also be influenced by protein degradation, right?

-fig 4 should be a mix also with fig 6S, as there is an extensive discussion in the text about the comparison of these 2.

-335, why not to make the experiment longer to be more precise?

-in the auxotroph experiment it would be good to know the amount of amino acid being exchanged, can this device be used for that?

-345-348. The authors repeat in the ms that they found a conversion from bidirectional (in well-mixed cultures) to unidirectional. However, they do not prove this. Probably in well-mixed cultures the population of meeting is also being dividing very slow and the % of these cells are lower than the other, which will fit their definition of unidirectional, however, bidirectionality is clearly required otherwise the met cells would not be able to supply the amino acid required for growing the other cells.

A major flaw of this work, in general, is the lack of control of well-mixed population, maybe just having in one chamber both populations or comparing with well-mixed population in flask experiments. This should be added.

-some conclusions are rather obvious like that growth in auxotrophic combinations is highly dependent on amino acid availability. Maybe they may be described in a way that you state that this was expected or obvious.

-fig5G. why do the authors expect those behaviors? Is there any evidence that says that both populations should grow at the same rate? Or that they both need and get the exact amount needed to have the maximum or the same growth? If not, any combination could be expected. For the observed(center) I still believe some bidirectionality exists, otherwise, there would be no growth at all of met mutant.

-400-405. This is the most interesting part and indicates that the metabolic state of the met changes if it is starving and provides more nutrients.. this should be validated by analyzing aa secreted in MM (minimum media) and MM+met. It would be a good addition to the ms, if more knowledge could be added in what is going on here at the molecular/metabolic level. Again, a control here with well-mixed populations would be very valuable.

- there are many ways to define spatial separation and this device present one way. However, I would expect that if you change the setup, there will be changes in the diffusion rates that can vary all the specific conclusions of this work and the distances required to see some effects may change.

-the autocad files must be presented as supplementary material, for other people to reproduce the device.

Reviewer #1 (Remarks to the Author):

*The authors have developed a microfluidic system that enables elucidation of molecular communication between cells in controlled microenvironments. The fluidic system provides for signal molecule transport across defined distances bounded by cells with different genetic backgrounds. The channels are constructed with a 0.5 micron height, which should severely restrict cell transport. The cells under investigation are engineered to (i) express fluorescent marker proteins in response to molecular stimuli (arabinose and 3-oxo-C6-AHL), or (ii) grow differentially in response to exogenously added amino acids. In addition, for the former, the authors have developed a relatively simple mathematical model that describes the expression of fluorescent markers within the engineered cells as a function of time. They have referred to the complete system as “MISTiC” for Mapping Interactions across Space and Time in Communities. In this way, the combination of the model and the microfluidic device enables discrimination of the signal transduction processes known to affect cell phenotype, which are particularly important for understanding behaviors within microbial consortia. Also, they have used the AHL signaling mechanism of *Vibrio fischeri*, which has essentially no exogenous impact on *E. coli*, the model system of choice.*

The manuscript is expertly prepared and the experiments appear to have been performed with care, including mathematical, statistical, and graphic representation of the data. The synthesis of these methodologies represents a fine contribution to the field in that there are few reports that provide the systematic amalgamation of the potentially disparate components that comprise this report. That said, there are several reports from several groups that have performed similar studies that have not been recognized in the current manuscript. The accumulation of these reports and their direct relevance to the current report, point to the question of novelty and partially diminish this reviewer's enthusiasm for this manuscript as a contribution in Nature.

For example, there are many groups who have examined cell signaling, including bacterial cell-cell signaling and even quorum sensing, in carefully controlled geometries. Atencia et al. (Lab Chip, 2009) created a microfluidic palette that is used to create carefully controlled chemical gradients in the absence of convective flows; they show how their system can be used to interrogate chemotaxis in response to carefully controlled glucose gradients. Jayaraman and coworkers (Lab Chip, 2012; J. Bact., 2014; Biomed Microdevices, 2014) have published an array of devices that enable carefully controlled microenvironments for the study of biological phenotype, including responses to chemical gradients. Diao, et al. (Lab Chip, 2006) constructed a system that interrogates bacterial responses that overlap with quorum sensing. Bentley and coworkers (Biomicrofluidics, 2017; Biotechniques, 2017, Lab Chip, 2015; Biomaterials, 2012) developed methodologies that demonstrate quorum sensing (QS) interrogation and control in similar systems. Correspondingly, they have developed agent based models to interrogate QS behaviors as a function of the molecular mechanisms of QS signal transduction (PLoS Comp Biol, 2016). These are a few of the many examples. Some of these methodologies reveal single cell behaviors, some reveal expression patterns at nearly single cell level, all provide visual basis for interpreting the phenotype in response to carefully controlled gradients. None of these reports, however, put all of these together in a way shown in the current manuscript.

We are encouraged that the reviewer recognizes our contributions to understanding diffusion-mediated microbial interactions in spatially structured communities. We thank the reviewer for bringing these papers to our attention. The majority of the papers developed different types of microfluidic devices to establish defined and tunable chemical gradients including Diao et al. Lab Chip 2006, Atencia J et al. Lab Chip 2009, Shang W et al. Biomicrofluidics 2017, Wang H et al.

Biomed Microdevices 2014, Kim et al. Lab Chip 2012 and Pasupuleti S et al. J Bacteriol. 2014. These studies used the gradient devices to study bacterial chemotaxis (Pasupuleti S et al. J Bacteriol. 2014, Shang W et al. Biomicrofluidics 2017, Diao et al. Lab Chip 2006 and Atencia J et al. Lab Chip 2009) or cellular responses to drugs (Kim et al. Lab Chip 2012 and Wang H et al. Biomed Microdevices 2014). We demonstrated distinct chemical gradients using a fluorescent dye in the MISTiC device for basic characterization (**Fig. S1**). However, the major focus of our paper is to precisely quantify interactions between bacterial strains physically separated by defined distances. In contrast to a precisely controlled chemical gradient, cells confined to the growth chambers produce and degrade molecules to establish a biologically generated gradient that could be *changing as a function of time* in response to environmental stimuli.

One of the studies described an image analysis method for bacterial chemotaxis (Pottash AE et al. Biotechniques 2017) and the other constructed an agent-based model to study cell-to-cell heterogeneity and pattern formation of genes regulated by quorum sensing (Quan DN et al. PLOS Comp Biol 2016). The Pottash AE et al. Biotechniques 2017 paper was particularly not relevant to our study since MISTiC was not intended for establishing chemical gradients for investigating bacterial chemotaxis. In the Quan DN et al. PLOS Comp Biol 2016 paper, cell-to-cell heterogeneity in quorum-sensing regulated gene expression has implications for pattern formation/spatial variation in gene expression. However, our study primarily focused on the role of defined spatial distance on gene expression and growth at the population-level and thus this paper was not particularly relevant. Examining how spatial separation impacts quorum-sensing regulated gene expression at the single-cell level is beyond the scope of the study and will be investigated in the future.

Finally, Luo X et al. Biomaterials 2012 and Luo X et al. Lab Chip 2015 used biofabrication of hydrogels to physically separate bacterial populations for studying quorum sensing bacterial cell-cell signaling. The MISTiC platform allows modulation of the distance between physically separated bacterial populations to study bacterial phenotypes in real-time. One potential limitation is the reproducibility of the exact size of the alginate hydrogels due to challenges with device fabrication. As such, variation in the porosity of the alginate hydrogels could alter the rate of diffusion of molecules between physically separated cell populations. In the Luo X et al. Lab Chip 2015 paper, media flows from one microfluidic device to the other and thus communication between cell populations does not occur by passive diffusion. We have now extensively revised our introduction to reference several of these relevant papers.

There were a few issues, however, that I feel should be addressed in addition to the issue of novelty. The first has to do with the most basic premise of the paper. The current manuscript emphasizes the distance between cells, yet recognizes that it is the concentration of the signal molecules that determine the fate of the biological response. Fixing this issue is relatively simple, but as currently prepared, the manuscript promulgates a suggestion that distances matter. This is only the case when convection, diffusion, cell migration, signal generation, and signal uptake are all controlled and static. Another relatively simple issue has to do with the use of diffusion rate and the systematic absence of diffusivity and a Fickian diffusion coefficient. The data could be recast as a function of the expected results given the diffusion coefficient and quiescent flow. The studies on fluorescein for example, could be used to abstract a diffusion coefficient, that in turn, could be used to validate that there is no convective flow in the device. Without this, there are limited control experiments that provide this insight. How do we know there was no convection? Pressure swings in these devices are common and difficult to control.

Along these lines, were studies done that demonstrate the absence of cell migration or motility? For example, if sterile media were provided on one side and cells and growth media on the other, would the sterile side stay sterile for the duration of the experiments here?

The reviewer is correct that convection and cell motility are important processes/cellular functions shaping microbial communities. Our device was designed to control for many of these factors and to only allow diffusion of biomolecules between the interacting growth chambers. We have revised the main text to indicate that our experimental and computational results focus on diffusion-mediated interactions in microbial communities in the absence of convection and cell motility.

The reviewer expressed concerns about convective flow between growth chambers. Our device was designed to minimize any pressure-driven flow between growth chambers. Our design is very similar to the diffusive gradient generator (Atencia et al. Lab Chip, 2009) that decouples convection and diffusion by balancing pressures at either side of a diffusion channel. In our device, the pressures between interacting growth chambers should be balanced because the flow rates, channel resistances, and outlet pressures are matched. Furthermore, the resistance of flow through the interaction channel is at least four orders of magnitude higher than the resistance of flow through the main channel based on cross-sectional area and length of these channels. We carried out new forced oscillator experiments where the position of the sender strain (GFP labeled) was switched in the MISTiC device. We compared the GFP fluorescence intensity to a previous experiment where the GFP-labeled sender strain was loaded into the opposite growth chamber. Our results showed that the position of the GFP-labeled sender strain did not alter the GFP fluorescence intensity (**Fig. R1**). In the presence of convection, we would expect to observe substantial differences in the average steady-state fluorescence intensity between growth chambers on opposite sides of the device and our results show that this is not the case. In sum, convection does not contribute to the microbial phenotypes that are quantified using MISTiC. We have included more discussion of convection in the updated manuscript.

The reviewer also asked about controlling for cell motility. The *E. coli* strains used in our paper are not motile. The device was designed to physically separate cells on either side of the interaction channels. The openings in the interaction channel are too small to allow *E. coli* cells to pass through, and therefore only allow biomolecules to diffuse between growth chambers. We never observed cells crossing the interaction channels in properly bonded devices. We have added a comment clarifying that cells on each side of the device are physically isolated and cannot pass through the interaction channel.

The bacterial strains in each experiment are labeled with different fluorescent reporters. As such, we can monitor the physical separation of the differentially labeled bacterial strains in the device as a function of time using fluorescent time-lapse microscopy. We have now included representative videos of our time-lapse fluorescence microscopy data to show that physical separation of the strains in the paired growth chambers was maintained for the duration of the experiment.

In regards to how we modeled diffusion, our models do contain a Fickian diffusion constant (D). We modeled diffusion by dividing the interaction channel into volume elements and used ODEs to model transport between these elements. As the reviewer mentioned, it would be interesting to estimate the diffusion coefficient (D) from our dye experiments. However, our dye gradient data was taken at steady state, and the steady-state gradient does not depend on D according to Fick's second law. In theory, it would be possible to design an experiment that monitored dynamic changes in the dye gradient in response to changes in the boundary concentrations. This dynamic data could then be used to estimate D . However, our manuscript focuses on quantitative analysis of several different microbial consortia in MISTiC. As such, determining the fluorescent dye diffusion coefficient would not impact our conclusions about the synthetic microbial communities in our study.

Figure R1. Sender GFP fluorescence intensity as a function of time in response to a forced oscillatory arabinose input with a period of 1 hr. The sender strain was loaded into either the left or the right growth chamber. Gray shaded regions indicate periods where cells were exposed to 0.1% arabinose. Solid lines indicate the mean for all growth chambers for a given distance. The interaction channel lengths are not indicated here for simplicity, as they have no effect on sender gene expression. The shaded regions indicate one standard deviation from the mean.

dynamics.

We carried out new data analysis to investigate the time required for activation and decay of the GFP (sender) and RFP (receiver) fluorescent reporters for steady-state oscillations in the forced oscillator experiment with an input arabinose period of 2 hours (Experiment 2, **Table 1**). The inferred activation and decay response times were approximately 30-40 min for GFP and RFP (**Fig. S5c,d**). The doubling time of the cells in this experiment is approximately ~30 min in LB media. Therefore, the time required for the fluorescence intensity to decrease by 50% is similar to cell doubling times. Previous work has demonstrated that GFP is highly stable in *E. coli* (Anderson, JB et al. *Applied and Environmental Microbiology* 1998). Therefore, GFP will only decrease by protein dilution, which is consistent with the stable fluorescence of the quorum sensing step response experiment (Experiment 1, **Table 1**).

While the majority of proteins in *E. coli* are highly stable with half-lives of 5-20 hr (Maurizi, MR, *Experientia* 1992), the LasI protein has been shown to be degraded by the Lon protease in *P. aeruginosa*. The *lon* gene in *P. aeruginosa* has 69.6% identity to its homologue in *E. coli*, suggesting the possibility that LasI may also be targeted for degradation in *E. coli* (Takaya, A et al. *Journal of Bacteriology* 2008). Since the fluorescent reporters could have significantly longer half-life than the proteins in the circuit such as LasI, the rate of decay of the fluorescent reporters will dictate the decay response time in our experiments. In our experiments, the number of cells in the growth chambers are approximately constant for the duration of the experiment for two reasons: (1) they are confined to grow in a growth chamber; (2) the cells continue to grow and divide since they are maintained in exponential phase under continuous flow conditions. We have included new data showing that the number of cells in the growth chambers remains constant as

A third issue might be a result of reviewer misunderstanding. There are many experiments wherein the fluorescence is observed to decrease, and rapidly. Indeed, the degradation rate constants in the model for LasI and GFP are non-zero (and different). Some clarification is needed relative to the rapid loss of fluorescence observed in the square wave experiments (the loss seems to be nearly the same rate as the increase), while the simple "ON" experiments seem to have a very stable fluorescence. Isn't it well known that GFP is stable and less well known that LasI may be less stable? Independent reference relative to stability would help. The data suggest that these two proteins have significant turnover rates, which is not expected and not shown in control studies. Perhaps either the number of cells in the co-culture experiments might be changing in time (owing to dilution of the fluorophore in the presence of cell growth; there were no data on the numbers of sender or receiver cells) or that the degradation of the fluorophores is dramatically important and affected by undescribed factors. Please clarify these

a function of time in a representative experiment (mixed auxotroph) using single-cell image analysis techniques (**Fig. S11b**). We have now updated our main text to discuss the timescales of activation and decay for the forced oscillator experiment with a period of 2 hr and include the new supplementary figure showing these data (**Fig. S5c,d**). In sum, the reviewer is correct that the rate of increase and decrease of fluorescence is approximately the same (~30-40 min) for the forced oscillator experiment with a period of two hours and the decay response times can be explained by the dilution of the fluorescent reporters due to cell growth/division.

Finally, the auxotroph experiments seem to be of limited utility. Cells receive many signals from a secretome, they grow, and they otherwise respond. The effects of the mutations are not simply reversed by the addition of the amino acid. The metabolic consequences of these mutations are widespread within the given host (for example, tyrosine synthesis also needs pheA). The fundamental insight obtained by these experiments is not clear.

Our auxotroph results section has been extensively revised with new experimental and modeling results. We summarize these changes below:

- We carried out a new experiment to quantify single-cell growth in the mixed auxotroph by mixing $\Delta metA$ and $\Delta pheA$ cells in equal proportions and loading the community into the growth chambers (**Fig. 5**). To quantify the growth rates of the cells, we developed single-cell image analysis methods to infer the maximum growth rate of single-cells in a mixed auxotroph experiment.
- We have developed a new computational model of the amino-acid cross-feeding interactions that recapitulates the trends in the maximum growth rates of the $\Delta metA$ and $\Delta pheA$ across all physically separated experiments (**Fig. 5b,c,f** and Supplementary Information).
- We have carried out experiments to investigate the release rate of the amino acids in the producer strain's filtered supernatants (**Fig. 5g**). These data validate the $\Delta metA$ rescue experiment (Experiment 8, **Table 1**) by showing that the release rate of phenylalanine is inversely related to the concentration of supplemented methionine. By contrast, the release rate of methionine by $\Delta pheA$ exhibits the reverse trend and is only detected in the presence of high phenylalanine concentrations. These data highlight the differential context-dependency of amino acid release into the environment.

We believe that these major changes to the paper have significantly improved our understanding of the *E. coli* auxotroph interactions and community stability. First, the single-cell analysis of the mixed auxotroph experiment demonstrates that $\Delta metA$ and $\Delta pheA$ exhibit non-zero growth in mixed culture for a period of time in the absence of supplemented phenylalanine and methionine. However, $\Delta pheA$ maintained a substantial growth advantage compared to $\Delta metA$ for the duration of the experiment (**Fig. 5d**). In spite of the asymmetry in the growth benefits provided each partner strain, $\Delta pheA$ and $\Delta metA$ maintain coexistence in the growth chambers for the duration of the experiment (**Fig. S11b**). The network asymmetry inferred based on single-cell growth measurements is consistent with the population-level measurements in the physically separated experiment in media lacking phenylalanine and methionine. Finally, our data shows that a high fraction of the population exhibited non-zero growth for the majority of the experiment whereas a small fraction of each strain was actively dividing. As such, the single-cell analysis demonstrates that elongation of $\Delta metA$ and $\Delta pheA$ cells dominated over cell division in the mixed auxotroph experiment (**Fig. 5d** inset, **Fig. S11c**).

We developed a computational model to recapitulate our physically separated experiments. The model was able to recapitulate the trends across different conditions (**Fig.**

b,c,f). Accurate fits of the model to the data required that $\Delta metA$ and $\Delta pheA$ exhibit different minimal growth rates, corroborating the possibility that the gene deletions impacted metabolism/growth beyond biosynthesis of the specific amino acids that the strains are deficient in producing. Other possibilities include the difference in the metabolic costs of synthesizing the amino acids (phenylalanine is more costly to produce than methionine (Mee MM et al. *PNAS* 2014)) and/or trace levels of phenylalanine in the MISTiC device released from $\Delta metA$. In addition, the model captured the positive and negative coupling between growth rate and methionine and phenylalanine, respectively which we demonstrated via metabolite measurements in the supernatants of the producing strains (**Fig. 5g**).

The secretion rates of phenylalanine and methionine were characterized as a function of $\Delta metA$ and $\Delta pheA$ growth, respectively, by titrating a range of concentrations of the supplemented amino acid for each auxotroph. We demonstrate that secretion of phenylalanine is inversely related to $\Delta metA$ growth, whereas the opposite is true for methionine secretion by $\Delta pheA$. One potential explanation for the qualitative difference in the release rates of methionine and phenylalanine as a function of growth rate is that methionine and phenylalanine are used in 23 and 2 different reactions in *E. coli*, respectively (Keseler et al. *Nucleic Acids Research* 2017). As such, methionine is a critical resource for the cell that may more rapidly transformed into other metabolites and/or more tightly regulated than phenylalanine.

Furthermore, the reviewer is correct that deletion of *pheA* and *metA* have additional effects on cellular metabolism and growth beyond the synthesis of phenylalanine and methionine, respectively. As such, the effects of these mutations on cellular metabolism and growth are not entirely rescued by supplementation of the amino acids. PheA carries out the first step in the parallel biosynthetic pathway for the aromatic amino acids tyrosine and phenylalanine, as well as the second step in phenylalanine biosynthesis. MetA catalyzes the first step of de novo methionine biosynthesis as well as promotes tolerance to n-butanol and mediates changes in growth in response to heat shock. This further validates that $\Delta metA$ and $\Delta pheA$ have different minimal growth rates in the MISTiC device consistent with our model.

Amino acid cross-feeding is often suggested as a strategy to enhance strain coexistence and community stability (see Ziesack M et al. bioRxiv 2019, <https://doi.org/10.1101/426171> for example). However, our results highlight the sensitivity of the amino acid cross-feeding interactions to spatial, environmental and temporal perturbations. Together, we believe that these substantial experimental and computational efforts that were carried out in response to the reviewer's comments have significantly strengthened the auxotroph section of the paper.

There are a few relatively minor issues:

1. P. 5. What is D_2 ? Is there a way to estimate this?

The parameter D_2 is a lumped parameter to represent the diffusion rate of AHL from interaction channel into the growth chambers and from the growth chambers into the main channels as described in the main text and Supplementary Information. We estimated D_2 by fitting our quorum-sensing model to the time-series measurements of average fluorescence of GFP and RFP for each interaction channel length in the step-response experiment (251 min^{-1}).

2. P. 5. How many cells are there? 150-200? I thought the phase contrast images could provide this number. It is supposed to be constant in the first sets of experiments.

We now show the number of cells in the growth chambers as function of time for a representative experiment. There are ~150 cells in each growth chamber (**Fig. S11b**). The number of cells in each growth chamber is constant as a function of time in our experiments due to the physical

confinement of the cells in the growth chamber and the media flow through the main channel which washes away cells as they grow and divide.

3. P. 3 (chemical signaling, multispecies) & P. 14 (metabolic burden). Should be careful to cite original references.

We have checked each of our citations to make sure that we are citing original references.

4. S1. Why no cell number equations? The mutation experiments aren't modeled...reinforcing their limited value.

In the revised manuscript, we included a new model for the auxotroph experiment that recapitulates the growth rate trends across distance in the physically separated auxotroph experiments.

5. S3. It seems like degradation tags would have been appropriate for the oscillatory input experiments.

We considered this possibility but chose not to fuse the fluorescent reporters or other proteins in the circuit to degradation tags since *ssrA* degradation tags have been shown to enhance the stochastic variation in circuit dynamics, potentially due to competition for shared proteases and saturation of the proteases (Potvin-Trottier, L et al. *Nature* 2017; Prindle A et al. *Nature* 2014). Indeed, a previous study showed that removal of the degradation tags from the circuit components in the repressilator (three ring oscillator) improved the regularity and robustness of the oscillator gene expression response (Potvin-Trottier, L et al. *Nature* 2017; Elowitz MB and Leibler S *Nature* 2000). Finally, degradation tags would also reduce the average fluorescence intensities and the dynamic range of the fluorescence measurements.

6. Fig S6. When one looks at these data and the overlaid gray bars, it is hard to directly visualize a response to the chemical input. This is especially problematic at low times (<400 min) and longer times (>500 min). This was noted in the manuscript, but it seems problematic in this figure.

We have carried out additional experiments to repeat the forced oscillator experiment with an input arabinose period of one hour several times. Across several experiments, the sender fluorescence intensity (GFP) shows a clear oscillatory response to the forced arabinose input oscillations (Fig. 3e). Interestingly, the receiver fluorescence intensity (RFP) displays a temporally variable response (Fig. 3f). In response to the 1 hr forced oscillation input period, we show that the signal-to-noise ratio increases as a function of distance, indicating that distance can improve the fidelity of information transmission in synthetic communities for this input frequency with the trade-off of a reduced output gene expression level.

7. Fig S8. What happens at 300 min?

As described in the main text, we infer the maximum growth rate based on the dilution of stable fluorescent reporters due to cell division. In these experiments, cells are grown in the MISTiC device in the presence of IPTG to induce the expression of LacI-regulated fluorescent reporters for a period of time. At a specific time point after cells have grown to fill the growth chambers, the media is switched to alter the methionine/phenylalanine concentrations and remove IPTG. Since the fluorescent reporters are highly stable and decrease via cell division/growth, the maximum growth rate can be inferred from the rate of decay of the fluorescence intensity as a function of time. In all of these experiments, there is a ~80 min time lag following the media switch for the

fluorescence to decrease as a function of time presumably due to saturation of the fluorescent reporter and time for altering gene expression/network activity states in response to the nutrient environmental shift. In the control experiment (Experiment 5, **Table 1**), fluorescence remains approximately constant during the period of time following the media switch prior to fluorescence intensity decay, whereas in the methionine and phenylalanine dropout experiment (Experiment 6, **Table 1**) the fluorescence intensity increases immediately following the media switch. Thus, we can infer that in the dropout experiment, the media switch results in a substantial decrease in the growth rate of the strains, which therefore leads to the accumulation of fluorescent proteins and an increase in fluorescence intensity. We hypothesize that the fluorescence decays ~80 min following the media switch since protein production is halted and the remaining pools of intracellular fluorescent proteins decrease steadily due to cell growth and division.

8. Fig S10. Does the data in "a" make sense?

The minimum doubling times of the $\Delta metA$ strain in several experiments that had low or zero methionine post media switch were close to the length of the experiment following the media switch (>1000 min). Therefore, the inferred doubling times for the $\Delta metA$ should be considered as approximate. As such, the statistically significant difference in doubling times in **Fig. S13a** may be due to noise across biological replicates (growth chambers at a fixed distance) as opposed to a biologically significant trend in the data. Corroborating this hypothesis, there are no other statistically significant differences in the doubling times for the $\Delta metA$ strain between distances in this experiment, suggesting that distance ranging between 25 and 250 microns did not modify the slow doubling time of $\Delta metA$.

9. Provide the host cells and their origins or all experiments. I believe the knockouts are isogenic to the host.

We have now included a strain table that describes the details about the strain background, plasmids harbored by the strains and relevant references in the revised manuscript (**Table 2**).

Reviewer #2 (Remarks to the Author):

The manuscript presents a microfluidic device called MiSTiC that can be used to study how distance and diffusion affect the exchange of molecules between two bacteria populations. They tested the device with AHL as a molecule of exchange and with two population of auxotrophs that require each other to survive.

I have several questions/comments I would like the authors to respond to.

- MiSTiC is referred as a tool to study spatially structured environments, however, it seems that the only variable that can be studied is the distance of the channel, what basically varies the diffusion of the molecules from one chamber to the other. For example, if more AHL reaches the second chamber more fluorescence is found, something rather logic and expected. The same can be found in the exchange of amino acids. It is well known that diffusion is key in bacterial communities and these experiments verify that.

The MiSTiC platform simplifies diffusion-mediated interactions to an environment where all variables are fixed, to isolate the effects of separation distance on bacterial interactions and create highly controlled perturbations to the environment to map the temporal response. The reviewer is

indeed correct that distance impacts the gradient and the concentration of AHL that reaches the receiver growth chamber. If a higher concentration of AHL reaches the receiver chamber, we would expect that higher fluorescence would be observed. Although this is expected, we use a computational model to demonstrate that distance-dependent changes in gene expression will only occur in the linear regimes of the dose response curve. For example, if the biochemical parameters of the circuits are altered, the gene expression output of the receiver could remain constant even though the distance from the sender population is varied from 25 to 250 microns. Therefore, our model suggests a quantitative and generalizable result that the impact of spatial separation is dependent on the biochemical circuit parameters (specifically the binding affinity and the cooperativity of the dose response).

Using the sender-receiver consortium, we explore how distance impacts signaling information transmission in response to variations in the frequency of input signals. We show that signal-to-noise ratio can increase or decrease as a function of distance depending on the signal input frequency. Further, distance establishes a low-pass filter for signal communication. These results can be leveraged to engineer circuits that communicate with greater fidelity and to better understand the role of noise in shaping microbial communication in communities.

We also carried out new MISTiC experiments using an *E. coli* consortium that displays bidirectional communication and dual-feedbacks via orthogonal chemical signals. Our results demonstrate that the circuit topology and feedback loops are major determinants of gene expression dynamics across spatial distance. Specifically, our data indicates the distance between the coupled oscillators globally reduces the stability of the oscillations over time. However, the positive feedback loop enhances the robustness of the activator oscillations whereas the negative feedback loop increases the fragility of the system, leading to abrupt loss of gene expression oscillations. These results were not expected and illustrate the complex interactions between network topology and spatial arrangements of members of a community in determining community behaviors.

Finally, we use the MISTiC platform to understand a complex and context-dependent mutualism between two amino acid cross-feeding *E. coli* auxotrophs. Our results show an unexpected asymmetric interaction between a slowly growing strain $\Delta metA$, which provides a substantial benefit to a partner strain $\Delta pheA$ that is inversely coupled to the producer strain's $\Delta metA$ growth. The growth benefit provided by $\Delta pheA$ was only detectable for a window of time in mixed conditions. Indeed, a distance of 25 microns eliminated the moderate and transient growth benefit from the partner strain $\Delta pheA$. Interestingly, quantification of methionine produced by $\Delta pheA$ was correlated with the growth rate of the producer strain whereas phenylalanine exhibited the reverse trend. Amino acid cross-feeding is often suggested as a strategy to enhance strain coexistence and community stability (for example, see Ziesack M et al. bioRxiv 2019, <https://doi.org/10.1101/426171>). Our results suggest that these interactions are highly context-dependent, change over time due to potential feedback interactions, and thus may not function reliably in complex environments. In sum, we believe the paper includes extensive characterization of the complexity of a predicted mutualism between two *E. coli* mutants across different environmental conditions. While the specific growth rates measured in these conditions are may be impacted by experimental details (e.g. flow rates), we believe that several of our results are generalizable. For example, generalizable results include qualitatively distinct amino acid release profiles as a function of growth rate and the sensitivity of amino acid cross-feeding interactions to small spatial perturbations. We believe that these results will be of broad interest to the microbial ecology and systems biology communities.

-in the abstract, the authors state "Our findings suggest generalizable principles that could be exploited to program the spatiotemporal behaviors of microbial consortia." However, it is not

clear what are these principles and how they can be used, and importantly how they are new or unknown before.

We thank the reviewer for requesting clarification on the generalizable principles of our work. We list below the principles that can be applied more generally to understanding the role of spatial parameters in shaping microbial consortia wherein interactions are dominated by diffusion and not cell migration/physical contact/convection. We have now revised our abstract to specifically describe the design principles suggested by our study.

- **The quantitative features of the input-output dose response of a signaling/metabolic pathway are major determinants of the effect of spatial separation on diffusion-mediated interactions in a community.** Our bacterial communication results demonstrate that distance-dependent gene expression is observed in interacting bacterial populations when concentrations of the diffusible compound map to the linear regime of the input-output dose response curve. Further, we pinpoint the binding affinity and ultrasensitivity of the signaling/metabolic pathway as major parameters that determine the sensitivity of bacterial growth/metabolism/signaling to spatial separation (**Fig. 2**).
- **The network topology/feedback loops of signaling/metabolic pathway are major determinants of the effects of spatial separation on diffusion-mediated interactions in a community.** We show that the effect of spatial separation for microbial interactions mediated by diffusion depends on the community network topology, which has generalizable implications for understanding community interactions in spatially structured environments and the engineering of microbial communities for a broad spectrum of applications in agriculture, biotransformations, human health and the environment (**Fig. 4**).
- **The frequency of environmental fluctuations can dictate the qualitative effect of distance on information transmission in microbial communities.** In natural environments, cells confront temporally changeable and uncertain extracellular chemical and physical stimuli. We varied the frequency of the input signal to determine how distance impacts information transmission between members of a consortium. Previous theoretical work has suggested that spatial averaging can dampen stochastic bursts in gene expression (Erdmann et al. *Phys Rev Lett* 2009). To our knowledge, we are the first to show experimentally that distance can reduce or improve reliability of information transmission in bacterial communities based on the signal input frequency. This finding highlights the precise control of spatiotemporal parameters enabled by the MISTiC platform for studying bacterial interactions. Spatial separation also reduces the mean output expression level (gain of the signal). Therefore, our results indicate a potential trade-off between the gain of the system and the signal-to-noise ratio in response to environmental fluctuations of specific frequency ranges (**Fig. 3**).
- **Amino acid cross-feeding is complex and context-dependent and necessitates detailed quantitative characterization to predict which parameter regimes generate robust mutualistic interactions.** Our results demonstrate that cross-feeding of amino acids in microbial consortia in continuous flow environments are time-dependent, sensitive to small spatial perturbations and environmentally context-dependent. First, we show concentration of the released amino acid and thus the strength of the outgoing positive interaction varies as a function of supplemented amino acid. Further, the qualitative relationship between the supplemented amino acid and the released amino acid depends on the metabolic pathway. Second, changes in relative growth rates between the strains demonstrate that the network topology changes as a function of time. Third, we show that

in mixed conditions, the community exhibits a transient mutualism for a period of time whereas only a unidirectional interaction was inferred when the strains were separated by 25 microns or larger. Together, these data demonstrate that metabolic interactions mediated by amino acid cross-feeding are highly sensitive to amino acid availability, spatial separation and change as a function of time. Therefore, these data suggest a generalizable principle that amino acid cross-feeding may not be a robust mechanism of microbial cooperation in natural and engineered communities (**Fig. 5**). Our results motivate the detailed characterization of the release of metabolites mediating autotrophies in microbial communities and could improve the predictability of cross-feeding across different environments.

-I find the introduction rather short and lacking more description of previous microfluidic approaches to study communities.

We thank the reviewer for this comment. We have extensively revised our introduction as per the reviewer's request.

-the authors should study how changes in the flow rate (input media flow) affect diffusion and exchange in the device.

We thank the reviewer for this suggestion. The media flow rate is an example of an environmental parameter that may alter the gene expression and growth responses of bacteria in the MISTiC device. To address this question, we have carried out new experiments to investigate how changes in the media flow rate influence gene expression and bacterial signal communication in the device (**Fig. R2**). To this end, we performed a step-response experiment using the sender and receiver *E. coli* strains. The strains were introduced into the device and grown for a period of time at 37°C in the presence of 25 µg/mL chloramphenicol, 0.5% Tween 20, and 62.5 ng/mL anhydrotetracycline (aTc). After 2.5 hr, cells were induced with 0.1% arabinose at a media flow rate of 200 µl hr⁻¹ to turn on the expression of LuxI and GFP. After 5.2 hr, the media flow rate was decreased to 20 µl hr⁻¹ for 3.05 hr. Finally, the flow rate was set to 500 µl hr⁻¹ for 7.2 hr. We evaluated the gene expression steady-state 4.2, 2.5 and 6.2 hours following the 200, 20 and 500 µl hr⁻¹ flow rate switches, respectively.

The sender and receiver gene expression steady-states decreased from 20 µl hr⁻¹ to 200 µl hr⁻¹ and plateaued at 200 µl hr⁻¹, demonstrating that our flow rate is in a saturated regime of this nonlinear function (**Fig. R2a**). The receiver gene expression decreased from shortest to longest interaction channels at each flow rate, demonstrating that the qualitative variation in receiver gene expression was robust to variations in media flow rate (**Fig. R2b**). The increase in the sender and receiver gene expression steady-states at low flow rates could be due to reduced nutrient availability/enhanced resource competition leading to a reduction in growth rate, which reduces the dilution rate of the fluorescent reporter and thus leads to an increase in average fluorescence across the cell population. We selected 200 µl hr⁻¹ because small ~2-5 fold changes in the flow rates do not alter the gene expression and growth response and thus the main media channel approximate an infinite sink for the molecules released by cells in the growth chambers. We decided to use a fixed flow rate that approximates an infinite sink to study different synthetic communities as opposed to investigating and modeling how flow rate impacts cell behaviors. While the contribution of flow rates to the bacterial gene expression and growth response is

interesting and highlights an additional parameter that we can precisely control using the MISTiC platform, it is beyond the scope of our study.

-line80. The characterization with the fluorescent dye is not really well studied. What is the diffusion coefficient of that dye? why the variation of the fluorescence with the distance is not symmetric? Is the dye also degraded on time?

We are not able to infer the diffusion coefficient of fluorescein because the rate of diffusion is faster than our imaging capabilities. Nevertheless, we carried out the parameter estimation of our diffusion model using the steady-state fluorescence intensity of fluorescein (Fig. S1). We inferred the parameters for a model that represented linear decay of fluorescein (model I, Fig. R3a) and a second model that did not include this decay term (model II, Fig. R3b). The sum of the mean

Figure R2. Steady-state fluorescence of (a) sender and (b) receiver strains as a function of the media flow rate in the MISTiC device. The sender and receiver strains were introduced into the device, induced with 0.1% arabinose and an62.5 ng/ml aTc after 151 min and the flow rate was set to 200 $\mu\text{l hr}^{-1}$. Following 465 min, the flow rate was changed to 20 $\mu\text{l hr}^{-1}$. The flow rate was then set to 500 $\mu\text{l hr}^{-1}$ following 648 min. The 20, 200 and 500 $\mu\text{l hr}^{-1}$ steady-state gene expression were measured 252, 151 and 374 min post flow rate switches, respectively.

squared errors across distances was 56% lower for model II compared to model I, indicating that a model that captures the decay of fluorescein significantly improves the model fit to the data. The decay of fluorescein in the interaction channels could be due to photobleaching, molecule degradation and/or absorption into the PDMS.

-In relation to the comment above, why the device is not designed in a symmetrical manner? In figure 1 a, the flow in blue could go in the opposite direction creating a symmetric display. If that is changed, would this affect to the symmetry of the fluorescent dye?

This is an interesting suggestion. The device is currently designed with parallel flow in the main channels to minimize pressure differences between interacting growth chambers. Minimizing these pressure differences prevents convective flow between growth chambers and allows us to study the effect of molecular diffusion. Reversing the flow in one of the main channels would cause pressure differences between interacting growth chambers, and result in pressure-driven

flow through the interaction channel. This would significantly change the fluorescent dye profiles. Near the center of the device, the pressures would be roughly equal resulting in dye transport that is dominated by diffusion. This would look similar to our current dye profiles. Interaction channels that are closer to the dye inlet would display dye profiles with a higher overall fluorescence than the diffusion-driven profile. Oppositely, interaction channels further from the dye inlet would display a lower overall dye profile. It would be interesting to study the effects of both flow and diffusion in future work.

-it is also unclear what is the metabolic state of the cells in the chamber, what are the cells that are washed out? The daughter cells? Are you concentrating older cells in the chamber? Do they

Figure R3. Diffusion model fit to fluorescein gradient in MISTiC interaction channels. (a) Diffusion model with degradation. The sum of the mean squared errors across distances is 0.0132. (b) Diffusion model with no degradation. The sum of the mean squared errors across distances is 0.0297.

replicate in the same manner across the length of the chamber?

The dimensions of the growth chambers in the MISTiC device are 10 μm side by 50 μm in length. Multiple cells are loaded into each growth chamber at the start of the experiment. As cells grow and divide, cells push their immediate neighbors towards the main channel. There is movement of the cells to the left and right walls of the channel. Since the channel width is 10 μm (~ 10 times larger than an *E. coli* cell), older cells are not trapped at the top of the chamber furthest from the main channels. The cells that are washed out of the chamber could be young-pole daughter cells or the old-pole mother cells (Wang P et al. *Current Biology* 2010). The investigation of single-cell growth and replicative age is beyond the scope of our study.

The growth channel dimensions were selected to achieve two main objectives: (1) maintain a small population size of ~ 150 cells that could produce molecules (e.g. quorum sensing AHLs or amino acids) at a sufficiently high concentration to impact the gene expression and/or

growth of a different spatially separated strain; and (2) achieve an approximately uniform mass transfer of nutrients in media to cells in the growth chambers (Yang D et al. *Front Microbiol* 2018).

Figure R4. Scatter plot of the correlation between the growth rate and depth position of individual $\Delta metA$ ($n=165408$) and $\Delta pheA$ ($n=193970$) cells in a mixed community (Experiment 7, **Table 1**). Positions zero and one represent the outermost and deepest regions respectively. Solid bold lines show the simple linear regression between maximum growth rate and chamber position. The Pearson correlation coefficient is 0.0959 ($P < 0.05$) for $\Delta metA$ and 0.0769 ($P < 0.05$) for $\Delta pheA$.

However, we carried out new data analysis of the mixed auxotroph experiment (Experiment 7, **Table 1**) to determine the relationship between the position of the cells in the growth chambers and growth rate (**Fig. R4**). Our results show weak negative correlations for the $\Delta pheA$ and $\Delta metA$ strains wherein cells deepest in the chamber displayed a moderately higher maximum growth rate than cells closest to the main channel. This moderate difference in the growth rate could be due to small differences in the local concentrations of methionine and phenylalanine within the growth chamber and/or variation in mechanical forces in different regions of the growth chamber (Yang D et al. *Front Microbiol* 2018).

-130-131. This is basically due to the fact that less AHL is reaching the cells, something not very surprising as when you add less AHL to cells in other configurations (tubes, flasks, plate..) you also get a lower response. Also there you mention the response time as not altered, but this is depending on the time scale you consider and the limitations of the measuring

times.

-the model results seem rather expected and logical. How do the authors believe the model can be used to generate new insights in the system?

The reviewer is correct that changing the distance between growth chambers alters the concentration of diffusible molecule in the recipient growth chamber. While this is expected, we show that the concentration of the diffusible molecule can map to very different biological responses depending on the quantitative features of the input-output dose responses that dictate the phenotypic response (growth, metabolic activity, gene expression) and/or the network topology/feedback loops of the intracellular network that mediates the biological response.

Our model results yielded insights into the spatial regulation of gene expression for diffusion-mediated microbial interactions. We summarize two major conclusions below:

- **The quantitative features of the input-output dose response of a signaling/metabolic pathway are major determinants of the effect of spatial separation on diffusion-mediated interactions in a community.** Our bacterial communication results demonstrate that distance-dependent gene expression is observed in interacting bacterial populations when concentrations of the diffusible compound map to the linear regime of the input-output dose response curve. Further, we pinpoint the binding affinity and ultrasensitivity of the signaling/metabolic pathway as major parameters that determine the sensitivity of bacterial growth/metabolism/signaling to spatial separation (**Fig. 2**).
- **The network topology/feedback loops of signaling/metabolic pathway are major determinants of the effects of spatial separation on diffusion-mediated interactions**

in a community. We show that the effect of spatial separation for microbial interactions mediated by diffusion depends on the community network topology, which has generalizable implications for understanding community interactions in spatially structured environments and the engineering of microbial communities for a broad spectrum of applications in agriculture, biotransformations, human health and the environment (**Fig. 4**).

-198-200. Maybe trying other QS molecules can help to prove this point.

We have carried out new experiments that study a synthetic *E. coli* microbial community oscillator (Chen Y et al. *Science* 2016). In this community, the activator strain produces C4-homoserine lactone (C4-HSL) and the repressor strain produces 3-OHC14-HSL. The combined signaling mechanisms establish coupled positive and negative feedback loops and bidirectional communication to control population-level gene expression. We studied this community into the MISTiC platform to investigate how distance impacts the dynamic oscillatory behavior of the consortium. Our results show that the (1) maximum cross-correlation between the activator and receiver decrease as a function of distance between the strains, (2) the time lag corresponding to maximum correlation increases as a function of distance, (3) the number of peaks and thus the temporal stability of the oscillations in the activator and repressor remain constant and decrease as a function of distance, respectively, (4) the coefficient of variation of the inter-peak distance or phase drift is approximately constant or increases as a function of distance, respectively (**Fig. 4**). In sum, these data demonstrate that positive and negative feedback enhanced or reduced the stability of the oscillator to variations in distance, respectively. In addition, spatial separation diminishes the strength of coupling between the activator and repressor oscillators.

-it seems that the noise in RFP expression could come from the specific times of expression, folding and degradation. Are these parameters known for the specific RFP used in this study? Maybe you can give values and learn the minimum time required to obtain a good response.

We assume that the reviewer is asking about the signal-to-noise results for the forced oscillator with a period of 1 hour (Experiment 3, **Table 1**). We performed three experiments that study the bacterial signal communication between the sender and receiver strains: (1) arabinose step-response (**Fig. 1**, Experiment 1, **Table 1**), (2) forced oscillator with a period of 2 hr (**Fig. 3**, Experiment 2, **Table 1**) and (3) forced oscillator with a period of 1 hr (**Fig. 3**, Experiment 3, **Table 1**). The variation in RFP expression across biological replicates as a function of time was low in Experiments 1-2. In Experiment 3, RFP displayed larger variability between biological replicates and irregular temporal variation around the mean. Previous computational work has shown that transmission of signals is limited by a critical frequency that is determined by the circuit architecture (Tan C et al. *Biophysics Journal* 2007). Information transmission is impeded for input frequencies larger than this critical frequency threshold. Our results suggest that a period of 1 hr is close to the critical frequency such that the information transmission is partially corrupted, leading to enhanced variability in RFP expression. In sum, GFP and RFP exhibit low variability in expression in Experiments 1-2, which suggests that the elevated noise in RFP expression in Experiment 3 is due to limitations in frequency-signal transmission as opposed to an inherent property of RFP. GFP and RFP are highly stable and decrease through cell dilution. The maturation times for mRFP (~20 min) and sfGFP (~14 min) are similar (Balleza E et al. *Nat Methods* 2018), suggesting the difference in reporter maturation time is not a major contributor to the variability in gene expression. In addition, the activation and decay response times for RFP and GFP were similar in the force oscillator with a period of 2 hr (**Fig. S5c,d**), indicating that maturation and degradation rates were approximately the same for both proteins. Our data suggests that the critical frequency threshold for communication between small bacterial

populations of ~150 cells across a 25 micron distance or longer is between a period of 1 to 2 hours.

-289, would not be possible to quantify the cells being washed out? Or leaving the chamber? That might be more precise. Fluorescence decay can also be influenced by protein degradation, right?

The GFP/RFP fluorescent proteins used in this study are highly stable and decrease primarily through cell dilution as opposed to proteolytic degradation (Anderson, JB et al. *Applied and Environmental Microbiology* 1998). Further, a previous study used this method to study *E. coli* growth, which provides additional support for the fluorescent dilution method (Roostalu J et al. *BMC Microbiol* 2008).

To analyze the mixed auxotroph experiment (Experiment 7, **Table 1**), we developed new custom single-cell image analysis code to quantify single-cell growth rates. The strain doubling times inferred using the single-cell method were similar to the doubling times inferred using the fluorescence dilution method. For instance, using the single-cell analysis method, the minimum doubling times of $\Delta metA$ and $\Delta pheA$ in the presence of methionine/phenylalanine were 92 and 84 min, respectively (Experiment 6, **Table 1**). Using the fluorescence dilution method, the doubling times for $\Delta metA$ and $\Delta pheA$ were ~85 min in the presence of methionine and phenylalanine (Experiment 5, **Table 1**). In addition, in the absence of methionine/phenylalanine, the minimum doubling time for $\Delta pheA$ in the mixed auxotroph experiment (197 min, **Fig 5d**) was similar to the minimum doubling time inferred by the fluorescence dilution method in the spatially separated experiment for 25 microns (243 min, **Fig. 5b,c**). Therefore, the maximum growth rates inferred by the single-cell and population-level fluorescent dilution methods were similar in methionine/phenylalanine, validating the accuracy of the fluorescence dilution method.

-fig 4 should be a mix also with fig 6S, as there is an extensive discussion in the text about the comparison of these 2.

We thank the reviewer for this suggestion. We have extensively revised this figure (**Fig. 3**) and have now combined the results for the forced oscillators with periods of one and two hours into a single figure.

-335, why not to make the experiment longer to be more precise?

The doubling times of the $\Delta metA$ strain in the physically separated experiments in the absence of methionine and phenylalanine (Experiment 6, **Table 1**) are >1000 min, indicating that $\Delta metA$ displayed very slow growth in these conditions. We believe that extending our experimental measurement time would not provide significantly more insight into the system beyond the fact that $\Delta metA$ grows very slowly but remains metabolically active to release F into the environment when physically separated from $\Delta metA$ by 25 microns or larger.

-in the auxotroph experiment it would be good to know the amount of amino acid being exchanged, can this device be used for that?

It may be possible to infer the concentration of the secreted amino acids based on a biosensor activity response. Here, a biosensor that induces a response (e.g. transcriptional regulation of a fluorescent reporter) to the released amino acid would be characterized in the presence of different concentrations of amino acid in the MISTiC platform to construct a standard curve that maps extracellular amino acid concentration to fluorescence. In the linear regime of the dose

response, the concentration of extracellular amino acid could be accurately mapped to fluorescence intensity. This would be interesting but beyond the scope of our study.

An alternative approach would be to develop a computational model that represents amino acid biosynthesis, diffusion, uptake and cellular growth. The model parameters could be inferred from the time-series data to estimate the secreted concentration of phenylalanine and methionine. We constructed a simplified and semi-mechanistic model to determine if we could recapitulate the maximum growth rates of $\Delta metA$ and $\Delta pheA$ as a function of interaction channel length in the spatially separated auxotroph experiments (Experiments 5-6, 8, **Table 1**). Our model is a coarse-grained representation of amino acid production, diffusion, uptake and utilization that was fit to the maximum growth rates of the $\Delta metA$ and $\Delta pheA$ across different spatial separations. As such, the kinetic parameters are lumped parameters that combine multiple mechanisms. Therefore, the model predicted concentrations of secreted methionine and phenylalanine are not exact and should be interpreted as approximate. Future work (beyond the scope of the current study) will develop a stochastic and more detailed mechanistic model that can be trained on single-cell growth rates to predict quantitative features of the system including the concentration of secreted amino acids.

To address the reviewer's question about amino acid release, we also carried out new experiments to measure the concentration of phenylalanine or methionine in the supernatant of the producer strains (**Fig. 5g**). Our results show that the concentration of secreted phenylalanine per $\Delta metA$ biomass is inversely proportional to the concentration of supplemented methionine. Conversely, our results show the opposite trend for released methionine wherein the concentration of methionine was detectable only in conditions with saturating supplemented phenylalanine.

-345-348. The authors repeat in the ms that they found a conversion from bidirectional (in well-mixed cultures) to unidirectional. However, they do not prove this. Probably in well-mixed cultures the population of meeting is also being dividing very slow and the % of these cells are lower than the other, which will fit their definition of unidirectional, however, bidirectionality is clearly required otherwise the met cells would not be able to supply the amino acid required for growing the other cells.

We have carried out new experiments to quantify the single-cell growth rates of an intermixed population of $\Delta metA$ and $\Delta pheA$ strains in the MISTiC platform (**Fig. 5d,e**, **Fig. S11**). Our results show the interaction network changes as a function of time and the growth rates of both strains eventually converge to zero, indicating the long-term collapse of the community. Specifically,

- The community exhibits a transient period in which both $\Delta metA$ and $\Delta pheA$ strains exhibit non-zero growth, suggesting that the mutualism exists for a period of time.
- The growth rate of $\Delta pheA$ is higher than the growth rate of $\Delta metA$ for the duration of the experiment, indicating that the network topology is asymmetric. $\Delta pheA$ continues to grow for a longer period of time than $\Delta metA$ demonstrating that non-growing $\Delta metA$ can maintain metabolic activity and produce phenylalanine to sustain the growth of $\Delta pheA$.
- After ~800 min the growth rates of both strains converge to zero, indicating long-term community collapse.
- Although the interaction network is asymmetric, the two strains coexist in the growth chambers for the duration of the experiment.

In the physically separated methionine/phenylalanine dropout experiment (Experiment 6, **Table 1**), $\Delta metA$ does not display a distance-dependent change in growth rate, suggesting that the growth benefit from $\Delta pheA$ is abolished by a 25-micron separation. Specifically, the growth of

$\Delta metA$ is transient in time and abolished by small variations in spatial separation. We have extensively revised our manuscript with the new results and data interpretation.

A major flaw of this work, in general, is the lack of control of well-mixed population, maybe just having in one chamber both populations or comparing with well-mixed population in flask experiments. This should be added.

This was a good suggestion. As per the reviewer's request, we carried out an experiment to introduce the $\Delta metA$ and $\Delta pheA$ strains into the growth chambers (mixed auxotroph experiment). We developed custom scripts to determine the single-cell growth rates within each growth chamber. These results show that the interaction network changes as a function of time from a mutualism (bidirectional positive), to unidirectional (non-growing $\Delta metA$ continues to support the growth of $\Delta pheA$) and finally to community collapse wherein the growth rates of both $\Delta metA$ and $\Delta pheA$ approach. Previous work demonstrated that the $\Delta metA$ and $\Delta pheA$ strains grow robustly in a well-mixed batch culture in a flask (Mee MT et al. *Proc Natl Acad Sci* 2014). Similarly, we performed a batch co-culture experiment wherein both strains were mixed together in equal proportions in a microtiter plate (Fig. R5a). Absorbance at 600 nm and fluorescence was measured as a function of time. These data show that a batch co-culture of $\Delta metA$ and $\Delta pheA$ exhibits a robust growth response and the fluorescence data suggests co-existence of the strains in this condition (Fig. R5b).

-some conclusions are rather obvious like that growth in auxotrophic combinations is highly dependent on amino acid availability. Maybe they may be described in a way that you state that this was expected or obvious.

These data show that trace concentrations of methionine and phenylalanine can abolish the distance-dependent growth benefit, highlighting the sensitivity of the interaction network to amino acid availability. We have now moved these data to the supplement to make room in Fig. 5 for results from the mixed auxotroph experiment.

Figure R5. Bulk growth based on absorbance at 600 nm and fluorescence time-series measurements of the $\Delta metA$ and $\Delta pheA$ co-culture in a microtiter plate (batch culture) using a plate reader. (a) OD600 vs. time for the co-culture. (b) Fluorescence intensity of RFP ($\Delta pheA$) and GFP ($\Delta metA$) as a function of time.

-fig5G. why do the authors expect those behaviors? Is there any evidence that says that both populations should grow at the same rate? Or that they both need and get the exact amount needed to have the maximum or the same growth? If not, any combination could be expected. For the observed(center) I still believe some

bidirectionality exists, otherwise, there would be no growth at all of met mutant.

The previous version of this panel showed a qualitative representation of the predicted topology of the expected networks based on previous literature. The expected networks are based on our data (**Fig. R5a**) as well as previous work suggests a bidirectional positive interaction between $\Delta metA$ and $\Delta pheA$ in batch culture in minimal media lacking methionine and phenylalanine (Mee MT et al. *PNAS* 2014). We agree with the reviewer that we do not have quantitative expectations for the node size (representing the growth rate of the two strains separated by 25 microns) and width of the edges (representing the distance-dependent change in growth rate from a 25 to 250 micron separation). Therefore, we have removed the expected networks and now show only the inferred networks based on the MISTiC experiments.

-400-405. This is the most interesting part and indicates that the metabolic state of the met changes if it is starving and provides more nutrients.. this should be validated by analyzing aa secreted in MM (minimum media) and MM+met. It would be a good addition to the ms, if more knowledge could be added in what is going on here at the molecular/metabolic level. Again, a control here with well-mixed populations would be very valuable.

To validate the $\Delta metA$ rescue experiment (Experiment 8, **Table 1**), we carried out new experiments to measure the concentration phenylalanine and methionine in the producer strain's filtered supernatant. Specifically, we grew $\Delta metA$ or $\Delta pheA$ in a flask (batch culture) with different concentrations of supplemented methionine or phenylalanine for a period of time to reach exponential phase. We measured the cell density using absorbance at 600 nm and filtered the supernatant for metabolite quantification. Our results showed that the concentration of phenylalanine per unit grew $\Delta metA$ OD600 (normalized by cell density) was inversely related to the concentration of methionine, corroborating the results of the $\Delta metA$ rescue experiment. These data are shown in **Fig. 5g**.

We performed the reciprocal experiment with the $\Delta pheA$ strain wherein $\Delta pheA$ was grown in media supplemented with different phenylalanine concentrations. Our results show that in all conditions except the highest phenylalanine concentration, methionine was below the 0.5 μM detection limit of the assay. Together, the amino acid measurements support our findings by demonstrating that (1) the secretion rate of phenylalanine is inversely related to the $\Delta metA$ growth rate and (2) the $\Delta pheA$ release rate of methionine is very low regardless of the growth rate/supplemented phenylalanine concentration.

- there are many ways to define spatial separation and this device present one way. However, I would expect that if you change the setup, there will be changes in the diffusion rates that can vary all the specific conclusions of this work and the distances required to see some effects may change.

This is a good point and the reviewer is correct in recognizing that the specific conclusions and the spatial effects studied here will change with different systems, device geometries, and experimental conditions (concentrations, flow rates, etc.) However, we have tried to draw more general conclusions about how microbes interact based on these results. For example, in the auxotroph experiment, we found that $\Delta metA$ does not receive a distance-dependent growth benefit from $\Delta pheA$. We don't suggest this particular value is generally true for all flow rates, simply that the interaction from $\Delta pheA$ to $\Delta metA$ is sensitive to environmental conditions and spatial configurations. In the revised manuscript, we have added this point to the discussion and clarified which phenomena we believe may be general beyond our particular setup. Similarly, while distance dependent changes in gene expression in the sender-receiver consortium were detectable between 25 μm and 250 μm , changes in the dose response mapping AHL to receiver gene expression may yield no distance-dependent changes in output response over this same

spatial range. Our computational model provided this insight and was used to generalize our findings beyond the particular system studied here. More broadly, our work suggests that feedback loops and quantitative features of input-output dose responses can be modulated to program distance-dependent gene expression dynamics.

Thus, one of the strengths of this work is the use of MISTiC coupled to computational models to dissect the spatial sensitivity of diffusion-mediated microbial interactions across a variety of systems. Future versions of MISTiC can be fabricated with different distances, geometries, number of inputs, etc. to study a broad repertoire of microbial interactions. Even without altering our device design, we were able to quantitatively probe the dynamic behavior of three different synthetic *E. coli* communities and extract novel insights about these systems. We believe the framework presented can be applied to investigate the time-dependent interaction network and spatial sensitivity of diverse communities. These principles can be exploited to leverage spatial distance as a mechanism to program microbial interactions for biotechnological and biomedical applications.

-the autocad files must be presented as supplementary material, for other people to reproduce the device.

We have provided the autocad files for the MISTiC platform.

Reviewers' Comments:

Reviewer #1:

Remarks to the Author:

The authors have carefully considered this reviewer's concerns, provided new data, and revised their conclusions. They have also recast their work in the context of the literature, citing previous works that are relevant.

Reviewer #2:

Remarks to the Author:

The authors of this work have addressed many of the comments and carried out a good number of additional experiments, which I believe have improved the manuscript's conclusions and may help to understand better the potential and limitation of the MISTiC device.

My major concern is that the authors need to be more cautious on what they claim as new knowledge and the tone they use, as sometimes, the explanation to explain the observed results is described as something unknown, when is, in my opinion, something proven in other systems and rather logic and obvious many times (i.e. use more 'as expected' instead of 'unexpectedly/surprisingly').

Some examples: Replying one of my first questions the authors summarise what they claim as novel or new knowledge, I have some doubts about some of them, and I think they should be re-written accordingly across the main text, as some of them are general knowledge proven many times.

-They claim: "The quantitative features of the input-output dose response of a signalling/metabolic pathway are major determinants of the effect of spatial separation on diffusion-mediated interactions in a community." It is well known that concentration affects output, and distance, in this case, means diffusion and concentration. The authors found this link, which is great, but it is not something new. My major concern is the way this finding is described. It is not a new discovery.

-They claim: "The network topology/feedback loops of signaling/metabolic pathway are major determinants of the effects of spatial separation on diffusion-mediated interactions in a community." This is rather obvious. Different topologies react differently at different concentrations/distance. Why is this surprising? This is general knowledge and one of the reasons why synthetic biologists keep improving genetic circuits. The same conclusions are often found in flask in well mixed populations. Such network topologies are a good explanation to the results obtained but are not a new finding.

The authors keep using the term 'unexpected assymetry' to talk about the auxotroph experiment. I reiterate my previous comments. This is not at all unexpected. It would be unexpected to see perfect symmetry, as the production rate of different amino acids is different as well as their use by the other strain.

Regarding the diffusion coefficient of fluorescein, I think it can be calculated in a different system (larger), it does not need to be determined using MISTiC, as they claim they face the limitation of the imaging capabilities. A simple macroscopic setup could provide that value.

I appreciate the discussion about the topology of MISTiC when I suggested making it symmetric. The authors believe the topology would not affect the conclusions of the paper. A simple experiment to test this would be to invert the populations from the chambers and see if the behaviors of the strains/input/output remain the same or not. If the results are not the same..would this make the conclusions of the manuscript less general?

I am still not fully convinced on the term 'unidirectional'. The fact that one population is not growing does not mean they are not exchanging molecules and amino acids to, for example, produce new proteins and keep the cells metabolically active. I don't think the experiments demonstrate the conclusions of the authors, and the text should reflect this uncertainty unless it is demonstrated, for example with C13 labeling one of the strains.

Some of the figures included in the response to reviewers are not included in the new version of the manuscript. I think readers would benefit of those as well.

Fig R5B is very interesting. Why do the authors think in microplate the community does not collapse as it happens in MISTiC? I would expect to see a situation in MISTiC (certain length or fluxes) where the device mimics a well-mixed population. This would really boost the potential applications of MISTiC, otherwise only applicable to microfluidic systems with a very specific configuration.

The manuscript would benefit from a last section where other possible uses of MISTiC are discussed as well as the limitations of the device. This is a good place to also discuss the several things they have defined as 'out of the scope' while responding to the reviewers' comments.

Reviewer #2 (Remarks to the Author):

The authors of this work have addressed many of the comments and carried out a good number of additional experiments, which I believe have improved the manuscript's conclusions and may help to understand better the potential and limitation of the MISTiC device.

My major concern is that the authors need to be more cautious on what they claim as new knowledge and the tone they use, as sometimes, the explanation to explain the observed results is described as something unknown, when is, in my opinion, something proven in other systems and rather logic and obvious many times (i.e. use more 'as expected' instead of 'unexpectedly/surprisingly').

Upon examination of the previous version of the manuscript, we found an instance of “unexpectedly” (line 505) and one instance of “unexpected” (line 598). We did not use “surprising” or “surprisingly” in the previous or current version of our manuscript. We used “unexpected” to describe the *E. coli* auxotroph results based on the following results. The $\Delta metA$ - $\Delta pheA$ batch co-culture exhibits substantial growth in our batch culture data (**Fig. R5** in first Response to Reviewers document) and previous batch culture data (see Mee MT et al. PNAS 2014 for example). Our understanding of this system is that each auxotroph could be rescued by compounds released by the partner strain to maintain viability and growth. Based on these data, our initial hypothesis was that both strains displayed similar growth rates to maintain co-existence in the community. For example, if the growth rates of the strains differed by 100-fold, then co-existence would be difficult to maintain as well as the concentration of released metabolites by the low abundance strain could be below a critical threshold to sustain the growth of the partner strain, thus leading to eventual community collapse. Our spatially separated MISTiC results showed that the *magnitude* of the difference in the growth rates of $\Delta metA$ and $\Delta pheA$ was large in a spatially separated context since $\Delta pheA$ displayed a ~250 min doubling time and $\Delta metA$ grew very slowly with a ~1200 min doubling time (**Fig. 5b,c**). However, we understand there may be different interpretations of these data and we have revised our manuscript to remove any instances of unexpected. We have also revised our manuscript to highlight the specific findings of our study.

To gain a deeper understanding of the community dynamics in a well-mixed batch culture, we carried out a new experiment where $\Delta metA$ and $\Delta pheA$ were introduced at three different initial proportions (50% $\Delta metA$, 50% $\Delta pheA$, 90% $\Delta metA$, 10% $\Delta pheA$ and the reciprocal) into culture tubes (well-mixed batch culture) that were serially transferred over approximately 5 days (**Fig. S9**). In one case, the cultures were transferred every 24 hours and the community OD600 decreased with the passage number, signifying community collapse. In the second case, the community was transferred initially at a 24-hour interval and then allowed to grow for a 48 hr period after the second passage. In this condition, the community exhibited growth after the third passage. For each time point that showed measurable growth, we quantified the fraction of cells that were $\Delta metA$ or $\Delta pheA$ by fluorescent microscopy. All conditions converged to a $\Delta pheA$ dominated state irrespective of the initial strain proportions. In sum, our data showed that the stability of the $\Delta metA$ - $\Delta pheA$ community was sensitive to the passaging time, potentially due to a decrease in the cell density at each passage. The $\Delta pheA$ strain grew faster than $\Delta metA$, consistent with our MISTiC experiments. We have revised our manuscript to include these new results.

Some examples: Replying one of my first questions the authors summarise what they claim as novel or new knowledge, I have some doubts about some of them, and I think they should be re-

written accordingly across the main text, as some of them are general knowledge proven many times.

-They claim: “The quantitative features of the input-output dose response of a signalling/metabolic pathway are major determinants of the effect of spatial separation on diffusion-mediated interactions in a community.” It is well known that concentration affects output, and distance, in this case, means diffusion and concentration. The authors found this link, which is great, but it is not something new. My major concern is the way this finding is described. It is not a new discovery.

We have carefully revised our manuscript to check our claims about novelty. We believe that the reviewer is referring to the statements made in the first Response to Reviewers document. To address the reviewer’s question: “the authors state “Our findings suggest generalizable principles that could be exploited to program the spatiotemporal behaviors of microbial consortia.” However, it is not clear what are these principles and how they can be used, and importantly how they are new or unknown before unknown before” in the first Response to Reviewers document, we responded by “describing the design principles suggested by our study” that could generalize beyond the specific conditions/systems studied in our manuscript. However, we did not claim that all of these conclusions were novel, but simply that our results supported these generalizable conclusions.

We certainly agree that we did not discover that spatial separation will impact the concentration of a diffusible compound, which will in turn alter a biological response. However, we show that MISTiC can be used to characterize in quantitative detail the changes in growth or gene expression as a function of micron-scale spatial separation. We used the sender-receiver consortium to quantify how variations in micron-scale separation impacts the receiver gene expression dynamics. We constructed a computational model to represent the sender and receiver dynamics. We used the computational model to quantitatively probe how specific biochemical parameters including the binding affinity of LuxR its chemical inducer (K_{LuxR}) and/or to its target promoter (K_{RFP}) or the Hill coefficient of the LuxR-AHL dose response (n_{RFP}) alter the pattern of steady-state gene expression across distance (defined as distance-sensitivity). Our results (**Fig. 2**) show that variations in these parameters have a large impact on whether the receiver steady-state changes across distance. For example, a ten-fold decrease in K_{AHL} from ~100 to 1 transforms a distance-dependent gene expression pattern into a constant output across the 25-250 micron distance range. Therefore, we show specifically how biochemical parameters can be altered to maintain a constant gene expression output as distance is varied. While the specific values of the parameters will vary across different systems as the model structure and estimated parameters change, we demonstrate that MISTiC can be used to parameterize a mechanistic model and the model can be used to identify which parameters could be modulated to program the output response. This detailed and quantitative understanding enabled by computational modeling of the sender-receiver consortium in MISTiC could be useful for synthetic biology/engineering applications to specifically program a biological response to changing distance (which may be more easily manipulated in some cases than the biochemical parameters). In sum, our approach of combining precise experimental manipulation of spatial arrangements of populations using MISTiC with computational modeling can provide a framework for how to quantitatively manipulate the spatial and biochemical parameters of the system to program biological responses. Finally, by varying the concentration of diffusible mediators via interaction channel length, MISTiC can be used to map a dose-response of growth or gene expression to a partner strain in the connected growth chamber, which would not be possible in a well-mixed culture, which has implications for community stability and functions.

-They claim: “The network topology/feedback loops of signaling/metabolic pathway are major determinants of the effects of spatial separation on diffusion-mediated interactions in a community.” This is rather obvious. Different topologies react differently at different concentrations/distance. Why is this surprising? This is general knowledge and one of the reasons why synthetic biologists keep improving genetic circuits. The same conclusions are often found in flask in well mixed populations. Such network topologies are a good explanation to the results obtained but are not a new finding.

The quoted sentence was in our first Response to Reviewer document and not found in our manuscript. We provided a general response because the Reviewer had requested a list of conclusions that could generalize beyond our paper. In the manuscript, we provide more specific conclusions based on our data. We used a dual-feedback oscillator with negative and positive feedback loops to demonstrate that the activator and repressor strain exhibited opposing trends in the amplitude and coefficient of variation of inter-peak distance (phase drift) as a function of distance. In addition, the number of peaks abruptly decreased at 250 microns spatial separation in the receiver whereas the activator maintained oscillatory behavior at this distance. Finally, we showed that the maximum cross-correlation and the time-lag of the maximum cross-correlation decreased with the length of the interaction channel, demonstrating that the strength of the coupling/synchronization of the oscillators was diminished with distance. A major difference between the activator and repressor is the presence of a positive and negative feedback loop, respectively. Therefore, our results suggest specifically that *feedback* is a critical feature of a synthetic circuit that can enhance the sensitivity or robustness to variations in distance. We show that negative and positive feedback reduces or enhances the robustness of the oscillations to spatial separation. Negative feedback has been shown to reduce noise in gene expression across a population (see Becskei A and Serrano L *Nature* 2000 for example) and thus is associated with enhanced stability and robustness in biomolecular circuits. Positive feedback can establish bistability, ultrasensitivity (see Venturelli OS PNAS 2012 for example) and signal amplification as well as system instability (divergence from an equilibrium or undesired oscillatory behavior). Our results highlight that negative can have a *destabilizing effect* on oscillations whereas the positive feedback loop can enable the system to maintain oscillations as a function of distance. We have revised this section to describe the specific findings in our paper.

The authors keep using the term ‘unexpected asymmetry’ to talk about the auxotroph experiment. I reiterate my previous comments. This is not at all unexpected. It would be unexpected to see perfect symmetry, as the production rate of different amino acids is different as well as their use by the other strain.

The reviewer is correct that the term “unexpected asymmetry” has different interpretations and is too general. We used this term one time in the previous manuscript (line 598 in the previous version). We had used “unexpected” to describe the *E. coli* auxotroph results based on the following results. The $\Delta metA$ - $\Delta pheA$ batch co-culture exhibits substantial growth as a function of time based on our data (**Fig. R5** in first Response to Reviewers document and **Fig. S9**) and a previous study (see Mee MT et al. PNAS 2014 for example). Based on this data as well as our understanding of this system (specifically, that each auxotroph could be rescued by the metabolites released by the partner strain in minimal media lacking the amino acids that each strain is deficient in producing), our initial hypothesis was that both strains must exhibit roughly similar growth rates to maintain co-existence in the community and achieve a sufficiently high concentration of the released amino acid to rescue the growth of the partner strain. In our manuscript, we found that the *magnitude* of the difference in the growth rates of $\Delta metA$ and $\Delta pheA$ was larger than expected (~250 min doubling time for $\Delta pheA$ and ~1200 min doubling time for

$\Delta metA$ at 25 micron spatial separation) and this is why we had used the term “unexpected.” However, we understand there may be different interpretations of these data and we have revised our manuscript accordingly.

To gain a deeper understanding of the community dynamics in a well-mixed batch culture, we carried out a new experiment where $\Delta metA$ and $\Delta pheA$ were introduced at three different initial proportions (50% $\Delta metA$, 50% $\Delta pheA$, 90% $\Delta metA$, 10% $\Delta pheA$ and the reciprocal) into culture tubes (well-mixed batch culture) that were serially transferred over approximately 5 days (**Fig. S9**). In one case, the cultures were transferred every 24 hours and the community OD600 decreased with the passage number, signifying community collapse. In the second case, the community was transferred initially at a 24-hour interval and then allowed to grow for a 48 hr period after the second passage. In this condition, the community exhibited growth after the third passage. For each time point that showed measurable growth, we quantified the fraction of cells that were $\Delta metA$ or $\Delta pheA$ by fluorescent microscopy. All conditions converged to a $\Delta pheA$ dominated state irrespective of the initial strain proportions. In sum, our data showed that the stability of the $\Delta metA$ - $\Delta pheA$ community was sensitive to the passaging time, potentially due to a decrease in the cell density at each passage. The $\Delta pheA$ strain grew faster than $\Delta metA$, consistent with our MISTiC experiments. We have revised our manuscript to include these new results. We show that the $\Delta pheA$ strain grew at a faster rate than $\Delta metA$, consistent with our MISTiC experiments. Together, these data suggest that one reason why the community collapses is due to the continuous flow environment that washes released metabolites. We have revised our manuscript to include these new results. We have omitted “unexpected” and added a few sentences to clarify the insights revealed by MISTiC.

Regarding the diffusion coefficient of fluorescein, I think it can be calculated in a different system (larger), it does not need to be determined using MISTiC, as they claim they face the limitation of the imaging capabilities. A simple macroscopic setup could provide that value.

We thank the reviewer for the suggestion. However, the fluorescein diffusion coefficient has already been measured previously several times in previous studies (see Perale G et al. *J Biomed Nanotechnol* 2011; Galambos P and Forster FK *Micro Total Analysis Systems* 1998; Rani SA et al. *Antimicrob Agents Chemother* 2005 for example). The objective of our paper is to quantitatively characterize the effects of defined spatial separation on diffusion-mediated microbial interactions. Determining the diffusion coefficient of fluorescein is not related to the central focus of our paper. We carried out measurements of fluorescein to characterize the chemical gradients established across the interaction channel and determine if our diffusion model could represent these gradients but not to directly measure the diffusion constant of fluorescein (see **Fig. S1**).

I appreciate the discussion about the topology of MISTiC when I suggested making it symmetric. The authors believe the topology would not affect the conclusions of the paper. A simple experiment to test this would be to invert the populations from the chambers and see if the behaviors of the strains/input/output remain the same or not. If the results are not the same..would this make the conclusions of the manuscript less general?

We thank the reviewer for this suggestion. This is a good control experiment to test whether there are any features of the device that can create pressure imbalances or convective flow. In MISTiC, the pressures between interacting growth chambers should be balanced because the flow rates, channel resistances, and outlet pressures are matched.

We performed the experiment suggested by the reviewer by inverting the chamber positions of the sender (GFP) and receiver (RFP) populations compared to the strain configuration show in **Fig. 2**. This experiment was carried out using identical conditions to Experiment 1 in

Table 1. The GFP (sender strain) and RFP (receiver strain) exhibited similar gene expression dynamics in both configurations (**Fig. R6**). In both arrangements, the GFP steady-state was constant across separation distance whereas the steady-state RFP decreased as a function of distance. A moderate difference in the RFP steady-state was observed in the 250-micron condition between the two experiments. However, this moderate variation in RFP steady-state is within the error of biological replicates of the same arrangement across different experimental days. Indeed, pressure imbalances or convective flow would have a greater effect on the smaller separation distances, which was not observed in our data. In sum, our results demonstrate that that MISTiC has a symmetric topology and separation distance as opposed to strain configuration in the growth chambers is the driving factor driving variation in strain growth and gene expression dynamics.

Fig. R6. Sender and receiver gene expression dynamics in two configurations in MISTiC. (a) GFP fluorescence (sender) as a function of time for original (dashed line, normalized data shown in **Fig. 1c**) and inverted (solid line) arrangements. The induction time (orange line) for the experiments were aligned, and fluorescence intensities were normalized to allow for direct comparison between the experiments. Shaded regions represent one standard deviation from the mean. (b) RFP fluorescence over time in the receiver growth chambers for original (dashed line, **Fig. 1d**) and inverted (solid line) chamber positions.

I am still not fully convinced on the term 'unidirectional'. The fact that one population is not growing does not mean they are not exchanging molecules and amino acids to, for example, produce new proteins and keep the cells metabolically active. I don't think the experiments demonstrate the conclusions of the authors, and the text should reflect this uncertainty unless it is demonstrated, for example with C13 labeling one of the strains.

In our previous version of the manuscript, the term “unidirectional” is used only to refer to the sender-receiver quorum sensing interaction (Lines 63, 131 in the previous version of the manuscript). This term was not used to describe the auxotroph interactions in the manuscript. For the spatially separated *E. coli* auxotroph experiments, we defined an interaction as the fold change in the maximum growth rate across different interaction channel lengths. Based on this definition, the interaction network is a unidirectional positive interaction from $\Delta metA$ to $\Delta pheA$, since the growth rate of $\Delta metA$ was constant across interaction channel lengths (**Fig. 5h**, middle). The reviewer is correct that the cells could be exchanging metabolites or competing for extracellular resources (nutrients in the media) that impacts cellular processes that are not

observable and does not lead to a measurable change in growth rate. Since we cannot directly observe metabolite exchange in MISTiC, we define an interaction based on changes in growth rate across distance. In microbial ecology, interactions are defined in many different ways. For example, interactions were inferred in Venturelli OS et al. MSB 2018 by fitting a generalized Lotka-Volterra model to time-series absolute abundance data of monospecies and pairwise assemblages. In Ghimire S et al. *mSystems* 2020, an interaction was defined as the fold change in colony forming units (CFU) in co-culture compared to monoculture. Here we defined the interaction based on the fold change in maximum growth rate of the 25 to the 250-micron interaction channels. We have revised our text to be more specific about this definition.

Finally, we thank the reviewer for the suggestion of C13 labeling, which could provide insights into the molecular mechanism of the interaction. This experiment is beyond the scope of our study which used MISTiC to quantitatively understand how spatial separation and temporal perturbations impact microbial consortia.

Some of the figures included in the response to reviewers are not included in the new version of the manuscript. I think readers would benefit of those as well.

We thank the reviewer for this suggestion. We have now incorporated several figures from our response to reviewers in the updated manuscript. These include:

- Fig. R3 was incorporated into the manuscript in **Fig. S1**.
- Fig. R4 has been moved to **Fig. S12f**.
- Related to Fig. R5, we carried out a new batch culture experiment in test tubes to characterize the population dynamics of the $\Delta metA-\Delta pheA$ co-culture is now included as **Fig. S9**

Fig R5B is very interesting. Why do the authors think in microplate the community does not collapse as it happens in MISTiC? I would expect to see a situation in MISTiC (certain length or fluxes) where the device mimics a well-mixed population. This would really boost the potential applications of MISTiC, otherwise only applicable to microfluidic systems with a very specific configuration.

To address the reviewer's question, we carried out a new experiment to characterize the community dynamics of the $\Delta metA-\Delta pheA$ co-culture (**Fig. S9**). These data show that the community is able to maintain growth and stability when the passaging time was extended to 48 hr, but not in a second condition where the passaging time was maintained at 24 hr. Based on these data and our MISTiC results, we postulate that the growth and stability of the $\Delta metA-\Delta pheA$ community requires a threshold concentration of M and F in the environment and thus a critical density of metabolically active/viable $\Delta metA-\Delta pheA$ producer cells to maintain this concentration above a critical level. In the spatially separated MISTiC experiment, the population size of each strain is limited to a ~150 cells within the growth chamber and this population size of each strain is even smaller in the mixed experiment (**Fig. S12d**). In addition, the strains are cultured in a continuous flow environment, which washes away a fraction of the released amino acids. Finally, the strains were spatially separated by 25 microns or more, which further reduces the concentration of the released amino acid in the partner strain's growth chamber. Future work will vary the media flow rate to assess how this parameter impacts community stability and growth of the $\Delta metA-\Delta pheA$ consortium to determine if specific flow rate regimes rescue the long-term stability of the community. In future work, we will mimic flow conditions of natural microenvironments, continuous culture (chemostats) and batch culture. However, this is beyond the scope of our current study which highlights features of three different systems in MISTiC (sender-receiver, dual-feedback oscillator and an *E. coli* auxotroph community).

The manuscript would benefit from a last section where other possible uses of MISTiC are discussed as well as the limitations of the device. This is a good place to also discuss the several things they have defined as 'out of the scope' while responding to the reviewers' comments.

We expanded the discussion section to describe applications and limitations of MISTiC as a tool to study microbial consortia. We summarize the advantages and limitations below.

Advantages of MISTiC:

- We can interrogate the role of micron-level spatial separation on microbial interactions by measuring bacterial growth, gene expression, cell size, morphology and/or emergence of cellular states.
- Enables precise control of flow rates and temporal shifts in environmental conditions.
- Maintains interacting strains over long periods of time to investigate microbial interactions in spatially separated but interacting chambers. Maintaining co-existence between competing microbial strains can be difficult to achieve in batch culture and continuous culture (e.g. chemostat) due to resource competition, etc.
- Designed to study the effects of diffusion-mediated interactions, which are a driving force for microbial communities, in the absence of cell-to-cell physical contact, convection and motility.
- Can be used to map a dose-response of a strain to the net environmental impact of a partner strain via modulation of the distance, which changes the concentration of diffusible mediators.
- Precise control of environmental and temporal stimuli combined with time-series measurements of a biological response facilitates the development detailed computational models of microbial interactions.

Limitations of MISTiC:

- Other mechanisms of interaction including metabolite exchange cannot be directly observed. However, biosensors could be developed for specific metabolites of interest.
- Investigates pairwise interactions between two populations.
- The population size is uniform across the device.
- Designed to study diffusion-mediated interactions in the absence of physical contact, convection and motility.

Reviewers' Comments:

Reviewer #2:

Remarks to the Author:

The authors have addressed some of my comments and explained their position regarding the results.

I now understand the first hypothesis of the authors and why their findings were different from that. In this regard, I must say that their first hypothesis was not based on what is already known about communities. First natural communities, which exchange nutrients, are not based on the equal amount of cells of each member, they are mainly different. This is logical, as the exchange rate of different metabolites is different as well as the concentration of metabolites required for growth is also different depending on the metabolite itself. Therefore, the most expected outcome of a population is that the two members are found in different concentrations. This is widely accepted. I understand that the authors were not aware of this and their results made them realise that this is the case.

I thank the authors for the new experiment. Please include figure R6 in the manuscript as this can be informative for the readers. the authors mentioned that the variations in RFP are within the error, have they done statistics that support this? they must be included in the supplementary material and discussed accordingly.

Regarding the discussion on the terminology 'unidirectional', I think the authors should include in the text their definition, as they explain in the response to my comment that different authors define interactions in a different way it is important to be clear to avoid misunderstandings.

Reviewer #2 (Remarks to the Author):

The authors have addressed some of my comments and explained their position regarding the results.

I now understand the first hypothesis of the authors and why their findings were different from that. In this regard, I must say that their first hypothesis was not based on what is already known about communities. First natural communities, which exchange nutrients, are not based on the equal amount of cells of each member, they are mainly different. This is logical, as the exchange rate of different metabolites is different as well as the concentration of metabolites required for growth is also different depending on the metabolite itself. Therefore, the most expected outcome of a population is that the two members are found in different concentrations. This is widely accepted. I understand that the authors were not aware of this and their results made them realise that this is the case.

The reviewer is correct that it is logical to expect different consumption and release rates of metabolites in natural communities. As a consequence, the abundance of members of a consortium vary across a broad range. However, amino acids are required for biosynthesis of the proteome and intersect with numerous other pathways in the cell. Indeed, 2.9% (50,000 molecules per cell) and 3.5% (40,000 molecules per cell) of an *E. coli* cell is composed of methionine and phenylalanine (see Yuan J et al. *Nat Chem Biol* 2006 and Neidhardt F et al. *Physiology of the Bacterial Cell: A Molecular Approach*), respectively among all amino acids by weight. Amino acid biosynthesis imposes substantial energetic burden and methionine and phenylalanine are two of the most energetically costly amino acids to produce in *E. coli* (Kaleta C et al. *Biotechnol J* 2013). As such, amino acid biosynthesis is a tightly regulated biochemical process by numerous feedback loops that optimize the balance between biosynthesis rates for cellular growth and maintenance while minimizing energetic costs (Sander T et al. *Cell Systems* 2019).

Given this prior knowledge about the high concentration of methionine and phenylalanine required for growth and viability, as well as precise regulation of biosynthesis rates, we hypothesized that both $\Delta pheA$ and $\Delta metA$ need to be sufficiently abundant in the community in the absence of supplemented methionine and phenylalanine to enable the growth of the partner strain and thus community growth. For instance, if one strain's growth rate is too low and becomes a very small fraction of the community, the concentration of the amino acid released to the partner strain could fall below a threshold amount required for growth and the community could tend towards collapse. Our initial hypothesis did not mandate equivalent abundances of each strain, but rather posited that orders of magnitude difference in growth rate between the two strains would likely not support a stable consortium. This initial hypothesis that $\Delta pheA$ and $\Delta metA$ must exhibit growth rates that are not orders of magnitude different, is unique to this amino-acid cross-feeding system based on our prior knowledge and is not a general conclusion for all microbial interactions. Our hypothesis was also informed by initial batch co-culture experiments of $\Delta metA$ and $\Delta pheA$ auxotrophs (see **Fig. R5** in first Response to Reviewers document). The co-culture exhibited substantial growth, with fluorescence data suggesting the presence of both strains at steady-state. Since these findings were supported by the literature as well (Mee MT et al. *PNAS* 2014), we hypothesized that the magnitude of the growth rate difference was not very large. This hypothesis would not hold for microbial interactions mediated by a trace nutrient for growth, which would not need to be produced in high concentrations to support the growth of the partner strain. However, as the Reviewer points out, there could be other potential interpretations of this hypothesis and it was removed from the paper in the previous revision.

I thank the authors for the new experiment. Please include figure R6 in the manuscript as this can be informative for the readers. the authors mentioned that the variations in RFP are within the error, have they done statistics that support this? they must be included in the supplementary material and discussed accordingly.

Upon the reviewer's request, we have included **Fig. R6** from the second Response to Reviewers document in the supplementary as **Supplementary Fig. 4**. As per the Reviewer's request, we computed the P-values of the steady-state GFP or RFP fluorescence intensities in Experiment 1, **Table 1** and the inverted position experiment (Experiment 2, **Table 1**). Specifically, a two-tailed t-test demonstrated no statistically significant difference between GFP steady-states between the two experiments across all separation distances ($P > 0.05$) (**Supplementary Fig. 4a**). According to the statistical test, the RFP steady-states were not different across 25, 50, and 100 μm separations ($P > 0.05$). There was a statistically significant difference ($P = 0.01014$) in the RFP steady-states for 250 μm distance (**Supplementary Fig. 4b**). This moderate difference (less than 2-fold) in steady-state RFP intensities between the two experiments is likely not due to pressure imbalances or convection since these factors would have a greater impact across shorter distances than longer distances. There are several potential factors that could contribute to the moderate variation: slight differences in initial growth prior to loading in the MISTiC device or experimental set-up (e.g. device fabrication).

Regarding the discussion on the terminology 'unidirectional', I think the authors should include in the text their definition, as they explain in the response to my comment that different authors define interactions in a different way it is important to be clear to avoid misunderstandings.

We agree with the reviewer that the definitions of microbial interactions in the literature are not standardized and that we should provide our definition in the main text. Our previous version of the manuscript defined how the interaction was computed. Nevertheless, we updated our manuscript to provide additional information about how the interaction is deduced from the time-series data. While previous interaction inference methods evaluate the difference in mono-culture and co-culture growth parameters (see Hsu RH et al. Cell Systems 2019 or Venturelli OS et al. MSB 2018), microbial interactions within MISTiC can be evaluated as the change in growth rate (or other phenotype) as a function of distance from the partner strain in a single experiment.